# Bridging circuit modeling and signal analysis to understand the risk of crosstalk contamination in brain recordings

Maria F. Porto Cruz [1,2] ✉, Elena Zucchini [2], Maria Vomero [1,3], Aldo Pastore [2,4], Ioana G. Vasilaş [1,3], Emanuela Delfino [2,4], Michele Di Lauro [4], Maria Asplund [3,5,6], Luciano Fadiga [2,4] & Thomas Stieglitz [1,3,7] ✉

Advancements in the field of implantable neurotechnologies have enabled the integration of hundreds of microelectrodes on ultra-thin and flexible substrates. Besides implantable components, also connectors, headstages and cables have to comply with the high-count demand, resulting in a complex and compact chain with reduced line spacing and smaller safety margins. Here, we show that epicortical recordings acquired from anesthetized rat brains with a state-of-art neural acquisition system are undoubtedly compromised by crosstalk, with signal coherence maps exhibiting a strong dependency to the routing layout. A crosstalk back-correction algorithm is developed, allowing to infer on how signals would look like under a zero-crosstalk scenario. We found that signal coherence between closely routed channels effectively drops after correction, corroborating crosstalk contamination. Our work stresses the importance of validating recorded data against the routing layout as a crucial step of data quality control, helping to come closer to ground truth data.

The idea of creating direct communication pathways between humans and machines has been around for half a century. As a product of decades of interdisciplinary research, brain-computer interfaces are today envisioned as highly miniaturized implantable devices capable of establishing intimate information streams with thousands of neural cells[1–3]. Strategies employed by such devices include the use of advanced ultra-thin flexible substrates that reduce the mechanical mismatch to the soft brain tissue and the integration of high-density microelectrode arrays that can map the brain with spatial resolutions in the sub-millimeter range[4–8]. Combining structural biocompatibility with fine spatial localization results in higher signal-to-noise ratio (SNR) and bandwidth, which ultimately translates into higher information content[9,10].

Miniaturized and high-density implantable devices require not only a decrease in electrode diameter, but, importantly, the downsizing of all elements leading up to the amplifier, including interconnect lines on the arrays, connectors, adapters and cables[11]. The limits of manufacturing and assembly are thus tested, resulting in reduced line clearances and smaller safety margins. Hence, as signal quality expectations become higher (i.e. higher SNR and bandwidth), technical specifications become stricter.

This study aims to fill a gap in the literature on implications of densely routed interconnection lines, including miniaturized compact

[1]Laboratory for Biomedical Microtechnology, Department of Microsystems Engineering (IMTEK), University of Freiburg, 79110 Freiburg, Germany. [2]Department of Neuroscience and Rehabilitation, University of Ferrara, via Luigi Borsari 46, 44121 Ferrara, Italy. [3]BrainLinks-BrainTools Center, University of Freiburg, 79110 Freiburg, Germany. [4]Center for Translational Neurophysiology of Speech and Communication, Istituto Italiano di Tecnologia, via Fossato di Mortara 19, 44121 Ferrara, Italy. [5]Freiburg Institute for Advanced Studies (FRIAS), University of Freiburg, Freiburg, Germany. [6]Bernstein Center Freiburg, University of Freiburg, Freiburg, Germany. [7]Department of Microtechnology and Nanoscience, Chalmers University of Technology, Kemivägen 9, 41296 Gothenburg, Sweden. ✉e-mail: mfportocruz@gmail.com; thomas.stieglitz@imtek.uni-freiburg.de

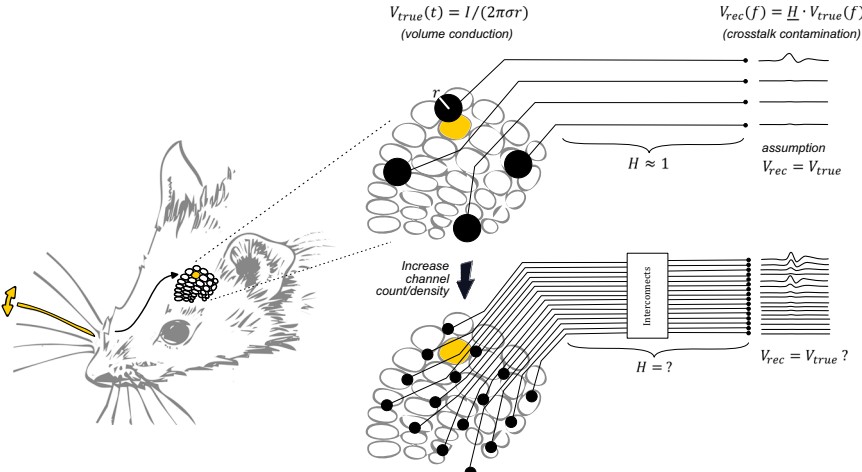

**Fig. 1 | Challenges in recording brain signals with high channel count arrays.**
Low channel count arrays allow recording signals from the brain surface when the active electrode is over or in close vicinity of the bioelectrical signals ($V_{true}$). These signals depend on the distance $r$ from the source to the electrode, its strength $I$ and the conductivity $\sigma$ of the brain tissue. The other electrodes are not able to record bioelectronic signals from the designated area and their interconnects most often run over "silent" areas (upper sketch). The transfer function (H) from the electrodes to the recording setup is nearly one and the recorded signals ($V_{rec}$) ideally reflect the true ones ($V_{true}$) (upper sketch). In high density and channel count electrode arrays, electrode size (diameter) as well as interconnection line width, pitch and insulation thickness decrease. Brain activity can be recorded with higher spatial selectivity. Since interconnection lines run over "active" areas of the brain and are very close together, either the brain signal can directly couple over the insulation material into interconnection lines or one line couples into an adjacent one. The transfer function H from the true brain signals to the recorded ones that reach the amplifier is not known in many cases. Recorded signals might not represent the ground truth which is distorted and convoluted by material and device characteristics (lower sketch). This study investigates opportunities to deconvolute recorded signals from material device properties by carefully characterizing each and every module a priori or a posteriori to a recording session and using post processing of signals to get closer to ground truth signals.

connectors required to connect microelectrodes to amplification stages, on the quality of in vivo data acquired with thin-film implants (Fig. 1). Electrochemical characterization of implantable electrodes, most often done by means of electrochemical impedance spectroscopy (EIS), has become a standard method of quality control, with studies typically reporting a boost in recording performance with decreasing electrode impedance[12]. Nonetheless, little is known on the effects of reduced insulation impedance (due to smaller safety margins) on recording quality. Indeed, as the interconnection lines conveying the recorded signals are more closely spaced, crosstalk, here intended as the undesired electrical coupling between a so-called active line and passive line, becomes more prominent[13].

Crosstalk in multielectrode intracortical electrodes has been previously proposed and evaluated in the literature using various circuit models, taking into account numerous design possibilities for recording or stimulation arrays. Starting from the initial idea of varying parameters such as track width and interelectrode distance[14], the ongoing need to increase electrode density for more precise mapping of the nervous system has led to the development of advanced electrical models. These models consider not only crosstalk between adjacent lines but also the effects of, for example, opposite traces in double-layer configurations[15,16]. The question is raised whether unwanted coupled signals can interfere with the true signals, as closely routed lines start outputting their own true signal along with a blended version of their neighbors' signal. What if, while aiming for finer spatial resolution, we are in fact losing discrimination between channels due to crosstalk? In a study from 2017, Nelson et al. investigated crosstalk contamination of action potential waveforms in recordings using glass pipette electrodes[17]. They concluded that crosstalk is not likely to affect data collected from neurophysiology experiments when both active and passive lines record similar magnitudes, although the study did not account for closely bundled interconnects sitting between electrode and amplifier, which are often the limiting factor. Most recently, Qiang et al. stressed the need to recognize electrical coupling as a potential hindrance to maximal signal quality in thin-film polymer microelectrode arrays and proposed a measuring platform for standardized and systematic characterization of crosstalk[18].

Here, for the first time, the bridge is made between crosstalk and its implications on the analysis and interpretation of continuous electrocorticography (ECoG) readings in rats, under all boundary conditions that are specific to a state-of-the-art neural acquisition setup featuring a polyimide-based conformable microelectrode array. Notably, the proposed updated version comes as a comprehensive system of the previously available equivalent electrical models and consists of six distinct blocks derived from an active and passive transmission line to emphasize the potential impact of individual blocks, encompassing both biotic and abiotic factors, on the reliability of the final recorded signal. Importantly, the assessment is done not only in the traditional local field potential (LFP) band, but also in the multi-unit-activity (MUA) band, only most recently explored in the ambit of microECoG readings[9,10]. This distinction is relevant, because crosstalk is a phenomenon that scales with frequency. Electrical coupling between active and passive lines occurs through the insulation between them and can be of resistive or capacitive nature. Resistive coupling refers to the physical contact between the lines (even though separated by an insulator) and is independent of frequency. Capacitive coupling arises from displacement currents generated by time-varying potentials along the lines and increases with frequency since the reactance decreases ($X_C = 1/wC$). In sum, the higher the frequency, the higher the expected crosstalk.

Signal coherence, defined as a measure of the degree of relationship between two time series as a function of frequency, is computed and analyzed with respect to the routing layout. This analysis allows to assess whether the relation between two channels depends purely on interelectrode distance at the cortex level, as it should, or if proximity at the routing level also plays a role. It is found that, for high frequency activity (above 300 Hz), coherence is highest for channels wired adjacently, even if the electrodes themselves are placed far apart on the cortex, thus strongly suggesting crosstalk contamination. To investigate this hypothesis, the recording chain from electrode to amplifier is fully characterized and modeled on the basis of impedance

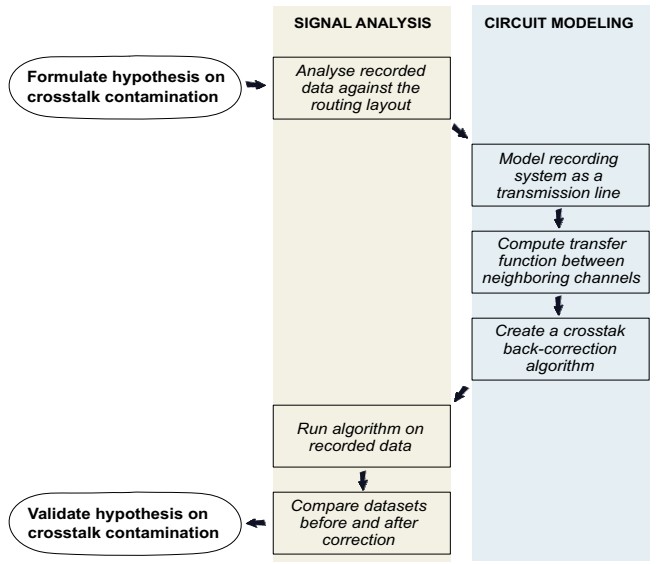

**Fig. 2 | Step-by-step approach employed in this work.** Hypothesis on crosstalk contamination is formulated by analyzing in vivo recorded data from rat brains against the routing layout of a state-of-the-art recording system. Following the routing layout, a lumped-element model is derived and the transfer function between neighboring channels computed for prediction of the crosstalk between them. A crosstalk back-correction algorithm original to this work is developed, allowing to infer on how signals would look like under a zero-crosstalk scenario. The recorded data is fed into the algorithm and signal characteristics before and after correction are compared to validate the hypothesis.

spectroscopy measurements, serving as a tool to simulate the crosstalk expected in the system in a step-by-step approach (Fig. 2). Then, in order to estimate its actual impact on the spectra of acquired ECoG signals, simulated coupling levels are used to develop an unprecedented algorithm for crosstalk back-correction of the recorded data. The algorithm effectively yields a decrease in coherence between closely wired channels, thus confirming crosstalk suspicions.

The prevalent trend is to dramatically increase electrode count and density in neural implants, at the risk of hastily assuming its benefit before setting boundaries on what are acceptable electrical coupling levels between channels. To address the issue, this work provides general guidelines and methodologies to identify cues for crosstalk contamination in in vivo electrophysiological data. Coupling levels can vary tremendously, depending not only on the setup itself, but also on the tissue-electrode interface developed after implantation. Therefore, it is crucial to include crosstalk evaluation as part of data quality control, since resulting distortions may otherwise lead to spatial discrimination loss or, even worse, to data misrepresentation and its erroneous interpretation as valid electrophysiological information.

## Results

An important clarification should be made on the terminology used in this paper. The term 'reference' (as in reference electrode) refers to the channel against which correlation and coherence maps are computed. Only in the methodology section, under specifically contextualized subsections, is the term 'reference' used to differentiate reference and ground electrodes from a recording configuration standpoint (in this study, these are shorted as unipolar recordings are used) or to differentiate reference, counter and working electrodes in an electrochemical setup. Placement of reference and ground electrodes significantly influences interferences (i.e. capacitive and/or inductive coupling from external sources) into the whole experimental setup but does not contribute to the effects that were investigated in this study.

## Discerning crosstalk from volume conduction

Electrocorticography (ECoG) maps of brain activity are analyzed and interpreted under the premise that signals originating in deeper cortical layers travel up to the epicortical surface by volume conduction. The spatio-temporally averaged potentials that reach the surface are picked up by the electrodes, resulting in signal amplitudes that are inversely proportional to the distance to the signal source[19]. Importantly, these signals should remain unchanged up to digitization to provide a realistic representation of the dynamics of the underlying cortical activity. However, as miniaturization of neural implants is pushed to the limit, with clearances between lines as small as a few micrometers, the question is raised whether the inevitable increase in crosstalk might result in distortion of signal transmission and, consequently, mis-representation of the spatial maps of activity (Fig. 1). It is important to highlight that the focus of this study is on the research question whether distortions in the electrophysiological recordings due to limited insulation resistance in the technical setup can be identified. The question of how volume conduction in biological tissues influences the signal shape and frequency component from the source to the transducer and how superposition of spatio-temporally distributed signal sources in such an environment works is not the aim of this investigation.

Here, somatosensory evoked potentials (SEP) are measured epidurally from the cortical barrel field of rat brains using a state-of-the-art conformable polyimide-based implant featuring a four by four array of platinum microelectrode disks, each with a 50 μm radius (Fig. 3a, b)[6]. The acquired signals are analyzed in two activity bands: the local field potential (LFP) band (3–300 Hz), typical of traditional ECoG monitoring, and the multi-unit activity (MUA) band (above 300 Hz), today accepted as part of the microECoG recording range.

The barrel C2, located among the four electrodes composing the top-left quadrant of the array (Fig. 3a), is activated by mechanical stimulation of the corresponding whisker and the resulting SEP and MUA spike waveforms are plotted for increasing interelectrode distance, taking electrode 1 (top-left electrode of the top-left quadrant) as reference. SEP waveforms show higher amplitude for the electrodes composing to the top-left quadrant, as expected due to closer proximity to the signal source, and progressively decay for increasingly distant electrodes (Fig. 3c). MUA activity follows the spatial distribution of the LFP activity: the higher the SEP amplitude, the higher the spike counts and spike amplitude, thus confirming that high-frequency signals collected from the surface of the cortex are directly linked to deeper cortical processes.

Nonetheless, when computing the spike cross-correlation between channels, taking electrode 1 as reference, it becomes visible that electrode 5, followed by electrode 9 and 13, have a cross-correlation higher than that predicted by spatial signal dispersion (Fig. 3d). To further explore the issue, signal correlation is computed for signals acquired from four arrays implanted on four animals, considering LPF and MUA bands separately. Plotting the correlation values for increasing interelectrode distance (taking electrode 1 as reference) confirms that correlation in the LFP band follows the principle of volume conduction, as correlation between electrodes decreases with increasing distance (Fig. 3e). However, the same plot for the MUA band shows that, once again, correlation to electrodes 5, 9 and 13 are consistently boosted with respect to the overall trend (Fig. 3f). According to the electrode arrangement (Fig. 3a), these three electrodes belong to the same column as electrode 1. In fact, the four electrodes composing each column of the array are wired adjacently throughout the recording chain. Thus, the correlation values strongly point to the hypothesis that they are, in fact, falsely increased due to crosstalk between neighboring channels.

## Recording system: from brain to amplifier

As a first step to understand how crosstalk affects the spectrum of the recorded signals, it is useful to estimate the electrical coupling

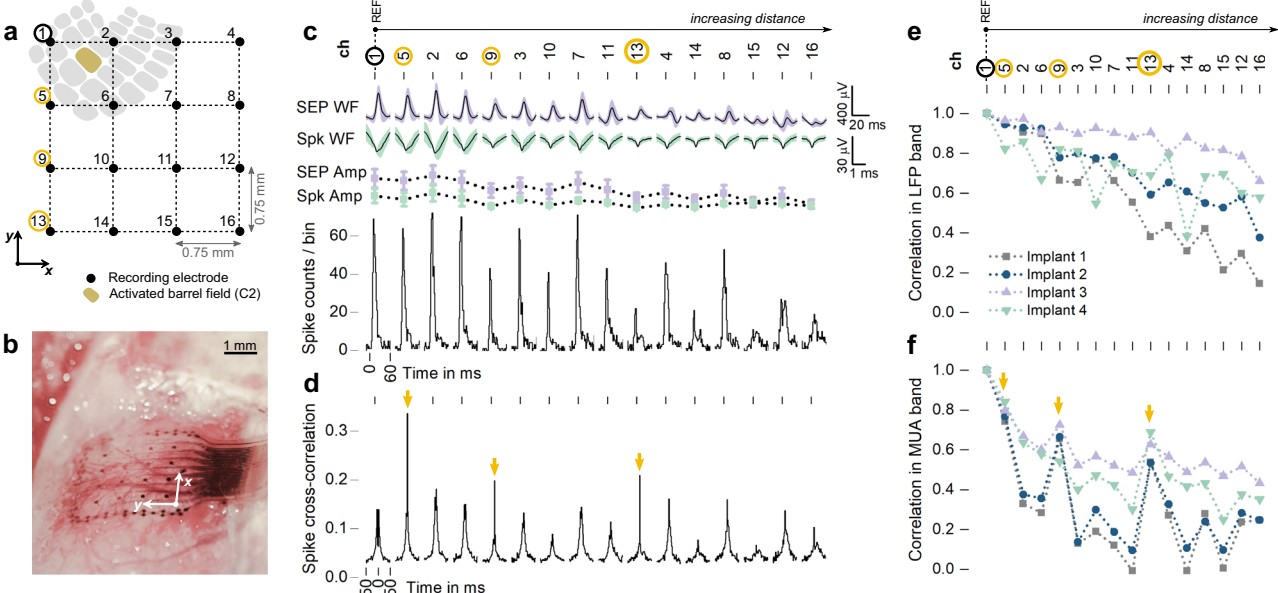

**Fig. 3 | Signal features according to electrode position. a** Schematic depicting the position of the four-by-four array of platinum microelectrode disks, each with a 50 μm radius, relative to the underlying barrel cortex (wS1) of the rat. The barrel field C2 is selectively activated following mechanical stimulation of the respective whisker. **b** Picture of the conformable polyimide-based implant, featuring the four-by-four electrode grid, wrapped around the rat's brain curvature. **c** Somatosensory evoked potential (SEP) waveform (WF) amplitudes, spike waveform (WF) amplitudes and spike counts, plotted for increasing interelectrode distance, taking electrode 1 as reference. Mean and standard deviation of SEP waveform amplitudes are calculated for $n = 60$ with n as the total number of stimulus repetitions for one implanted device. Mean and standard deviation of spike waveform amplitudes are calculated for all detected spikes for each channel across the full recorded response. **d** Spike cross-correlations computed using electrode 1 as reference. Reference spikes were not counted when calculating cross-correlogram versus itself. Electrodes 5, 9 and 13 have a higher cross-correlation than predicted by spatial signal dispersion. **e** Signal correlations in the LFP band, computed using electrode 1 of each array as reference for each of the four implanted devices (implanted on four different animals). **f** Signal correlations in the MUA band. Also here, electrodes 5, 9 and 13 are boosted in respect to the overall trend.

occurring along the recording chain, from brain to amplifier. ECoG recordings are here collected epidurally with a conformable polyimide-based implant (Fig. 3a, b) using a commercial Tucker Davis Technologies (TDT) multi-channel recording system. The polyimide thin-film is mounted on a male Omnetics-nano connector, which plugs into a female counterpart fixed on an adapter board. This intermediate board makes the connection to a ZIF-Clip® headstage, from which cables extend to the preamplifier, where signals are digitized. In Fig. 4, the recording system is simplified as two recording channels, one active (ACT) and one passive (PAS), and further broken down into six blocks.

The extracellular potential of the neuron is modeled as a voltage source $V_E$ and the tissue between source and electrode, which includes cortex and meninges, is simplified as a spread resistance $R_S$ (Block 1). In order to differentiate between the active and the passive line, the latter is not directly connected to $V_E$, but to ground, meaning it is silent. This is depicted by the gray (silent) versus purple (non-silent) color in the schematic of Fig. 4. The tissue-electrode interface is represented by the electrolyte resistance $R_E$, which accounts for the access resistance of the line, and the well-known Randles charge transfer element $R_F C_H$, consisting of a faradaic resistance in parallel with a capacitance representing the Helmholz double-layer (Block 2). The following two blocks refer to the implant itself (i.e. polyimide thin-film mounted on the male Omnetics connector), specifically the shunting path through insulation to ground, modeled as the element $R_{Sh} C_{Sh}$ (Block 3), and the coupling path between lines, represented by $R_{Imp} C_{Imp}$ (Block 4). After the implant comes the interconnects block combining all intermediate stages between implant and amplifier, i.e. female Omnetics connector, adapter board, ZIF-Clip® headstage and cables. The block is modeled as a coupling path $R_{Int} C_{Int}$ between lines (Block 5). Lastly, the amplifier's input impedance is represented as an element $R_{Amp} C_{Amp}$ shorted to ground (Block 6).

Each of the first five blocks of the lumped-element model are dimensioned according to specially devised impedance spectroscopy measurements, as summarized in the methodology section. The spread resistance $R_S$ is estimated as the access resistance extracted from an EIS measurement performed between an intracortical needle inserted 0.5 mm deep into the cortex and a surface epidural electrode ($Z_1$). $R_E$, $R_F$ and $C_H$ are fitted from the EIS curve of an implanted recording electrode ($Z_2$) and the shunting parameters $R_{Sh}$ and $C_{Sh}$ are estimated from the EIS curve of an insulated, i.e. non-functional, electrode submersed in phosphate buffered saline (PBS) solution ($Z_3$). The coupling parameters $R_{Imp}$ and $C_{Imp}$ are dimensioned according to the impedance measured between two neighboring channels of the implant ($Z_4$) and, correspondingly, the same measurement performed across the interconnection stage is used to estimate the coupling parameters $R_{Int}$ and $C_{Int}$ ($Z_5$). The values for $R_{Amp}$ and $C_{Amp}$, comprising the sixth block, are specified by the manufacturer ($Z_6$). All parameter values can be found in Table 1.

Crosstalk between active and passive lines occurs via two coupling paths: through the insulation at the implant level ($Z_4$); and through the insulation at the interconnects level ($Z_5$), which accounts for the female Omnetics connector, adapter board, ZIF-Clip® headstage and cables. The impedance curves (Fig. 5) show that the limiting coupling path (i.e. lowest impedance path) is via the interconnection stage, with the insulation impedance being one order of magnitude lower than that of the implant (5 MΩ vs 31 MΩ at 1 kHz). Adapter boards, headstages and cables (here generalized as interconnects), are often an underestimated element of the recording chain, which can indeed have a critical impact on the overall transfer function of the system.

### Simulating crosstalk

Breaking down the recording chain into blocks makes it easier to characterize the system and allows the impact of each block on the

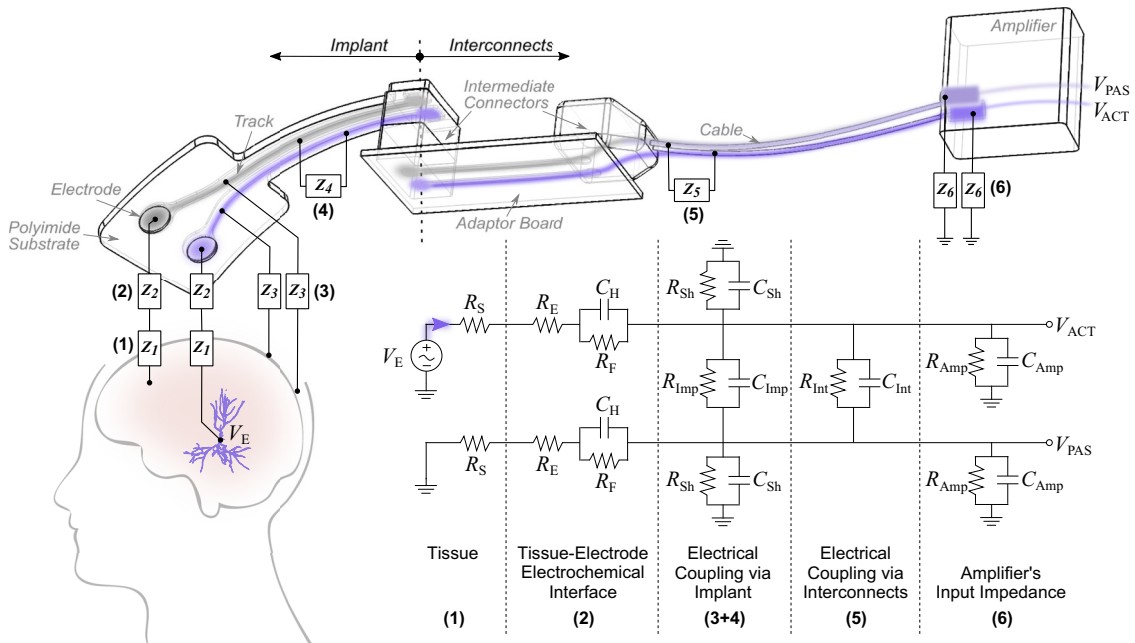

**Fig. 4 | Model of the recording system, from brain to amplifier.** The recording system is simplified as a transmission electrical model, featuring an active line (ACT) and a passive line (PAS). Each line is composed of a spread resistance ($Z_1$), tissue-electrode interface ($Z_2$), shunting path to electrolyte ($Z_3$), coupling path through implant insulation ($Z_4$), coupling path through interconnects insulation ($Z_5$) and the amplifier's input impedance ($Z_6$). Impedances are modeled as composition of resistors $R_i$ and capacitors $C_j$ with indices $i,j$ specifying it in more detail. The voltage source $V_E$ is considered as the extracellular potential of a neuron. Each element of the lumped-element model is defined in Table 1.

## Table 1 | Parametrization of the lumped-element model

| Block | Lumped-element | Symbol | Value | Unit | Extraction method | Physical meaning |
|---|---|---|---|---|---|---|
| 1 | Spread resistance | $R_S$ | 45.3 | kΩ | Access resistance | Brain tissue and meninges |
| 2 | Electrolyte resistance | $R_E$ | 7.6 | kΩ | Randles fitting | Electrolyte film wetting the cortex surface (accounts for the access resistance of the line) |
|  | Faradic resistance | $R_F$ | 12.0 | MΩ |  | Electrode-electrolyte interface |
|  | Helmholtz capacitance | $C_H$ | 4.4 | nF |  | Electrode-electrolyte interface |
| 3 | Shunt resistance to the electrolyte | $R_{Sh}$ | 0.6 | GΩ | RC fitting | Insulation between line and electrolyte through polyimide |
|  | Shunt capacitance to the electrolyte | $C_{Sh}$ | 23.0 | pF |  |  |
| 4 | Insulation resistance of the implant | $R_{Imp}$ | 1.6 | GΩ | RC fitting | Insulation between neighboring channels at the implant level (includes polyimide substrate mounted on male Omnetics nanoconnector) |
|  | Insulation capacitance of the implant | $C_{Imp}$ | 4.8 | pF |  |  |
| 5 | Insulation resistance of the interconnects | $R_{Int}$ | 12.4 | MΩ | RC fitting | Insulation between neighboring channels at the interconnects level (includes female Omnetics nanoconnector, ZIF-Clip headstage and cables) |
|  | Insulation capacitance of the interconnects | $C_{Int}$ | 43.9 | pF |  |  |
| 6 | Input resistance of the amplifier | $R_{Amp}$ | 2.5 | TΩ | Specified by supplier | Amplifier's input terminals |
|  | Input capacitance of the amplifier | $C_{Amp}$ | 8.8 | pF |  |  |

Description of all parameters used in the model illustrated in Fig. 4, according to the impedance spectroscopy curves shown in Fig. 5.

undesired coupled signal $V_{PAS}$ outputted by the passive line (Fig. 4) to be assessed individually. In simple words: how purple (i.e. non-silent) becomes $V_{PAS}$ after distortion through the coupling paths?

The transfer function between active and passive lines, simulated in Simulink using the lumped-element model, is shown as a black dashed line in Fig. 6. Represented as an attenuation magnitude, -∞ dB corresponds to zero coupling (maximum signal attenuation) and 0 dB to a hundred percent coupling between lines (no signal attenuation). Crosstalk is not constant throughout the recording band. In the LFP

band (up to 300 Hz), signal attenuation is relatively constant around -40 dB, while, in the MUA band, signal coupling significantly increases with frequency. From 1 kHz to 10 kHz, attenuation increases steeply from -34 dB (2% coupling in terms of signal amplitude) to -16 dB, which corresponds to a coupling of 16%.

Furthermore, the simulation predicts how the transfer function would change if each parameter would vary independently by a 5-fold and 10-fold increase and decrease (×5, ×10, ÷5, ÷10). Only the limiting parameter of each block is summarized in Fig. 6 (block 6 is excluded

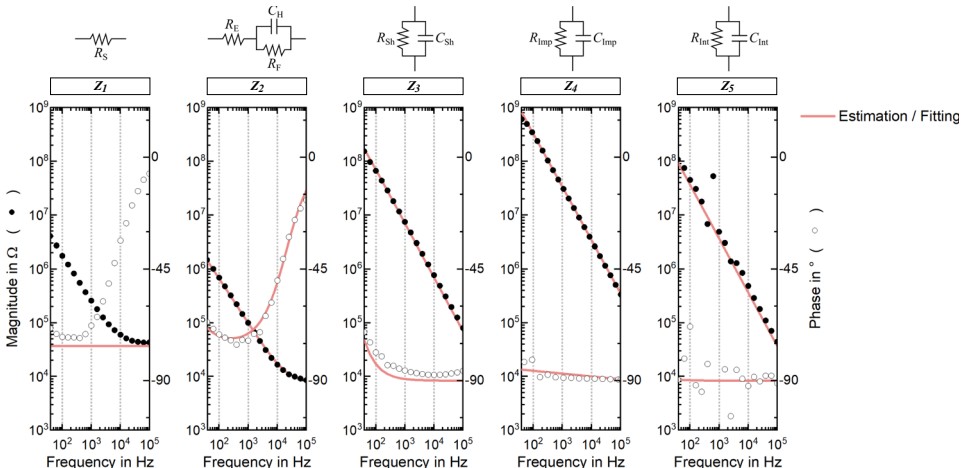

**Fig. 5 | Impedance spectroscopy for estimation of model parameters.** Each of the blocks ($Z_1$ to $Z_5$) composing the transmission model is dimensioned according to impedance spectroscopy measurements (except block 6, which is specified by the supplier). $R_S$ is estimated as the access resistance extracted from a measurement performed between an intracortical needle inserted 0.5 mm deep into the cortex and a surface epidural electrode. $R_E$, $R_F$ and $C_H$ are fitted from the impedance curve of an implanted recording electrode, according to the Randles model. The shunting parameters $R_{Sh}$ and $C_{Sh}$ are fitted from the impedance curve of an insulated electrode. The coupling parameters $R_{Imp}$ and $C_{Imp}$ are fitted from the impedance measured between two neighboring channels of the implant. The coupling parameters $R_{Int}$ and $C_{Int}$ are fitted from the impedance measured between two neighboring channels across the interconnection stage.

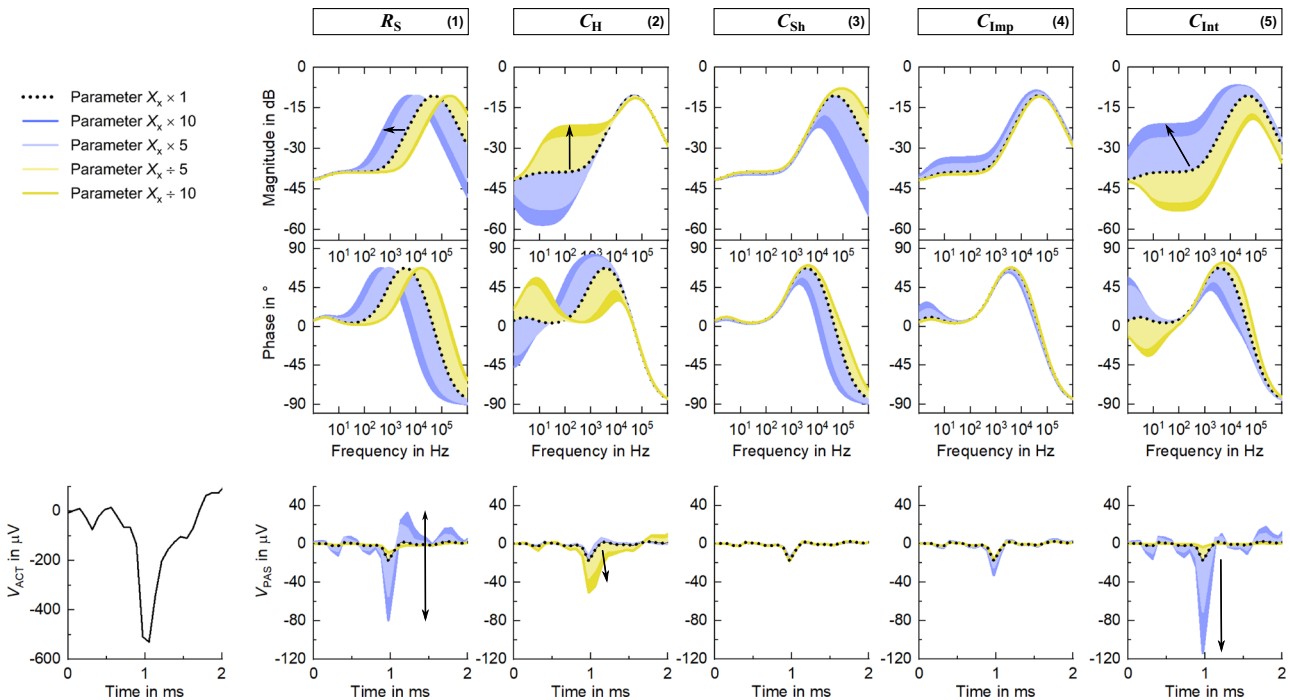

**Fig. 6 | Transfer function of the transmission model.** The baseline transfer function, assuming the parameters specified in Table 1, is represented by a black dashed line. The limiting parameter of each block (excluding block 6), i.e. $R_S$, $C_H$, $C_{Sh}$, $C_{Imp}$ and $C_{Int}$, is varied individually by a 5-fold and 10-fold increase and decrease (×10, ×5, ÷5, ÷10) and the resulting shift in the transfer function is colored in dark blue, light blue, light yellow and dark yellow, respectively. A spike waveform is fed into the active line and, for each parameter variation (×10, ×5, ÷5, ÷10), the resulting shift in the coupled signal is colored using the same color code, alongside the baseline coupled signal (black dashed line). In the bottom-left corner, the spike waveform registered by the active line is depicted.

here), which, for all blocks composed of both a resistive and capacitive component, corresponds to the capacitive part. Particularly, the key elements determining crosstalk are the spread resistance $R_S$, the double layer capacitance $C_H$ and the insulation capacitance of the interconnects $C_{Int}$. Increasing the resistance $R_S$ results in a shift of the transfer function towards lower frequencies, leading to critical crosstalk levels already in the LFP band. A decrease in capacitance $C_H$ is characterized by a boost in crosstalk in the range from 10 Hz to 10 kHz. For a 10-fold decrease, crosstalk reaches -20 dB at 1 kHz, which

corresponds to a staggering number of 10% signal coupling. Varying the capacitance $C_{Int}$ has a drastic impact in the whole frequency band, down to 1 Hz.

To understand how these shifts in transfer function translate into signal amplitude fluctuations, a spike waveform recorded extracellularly is used as the source signal injected into the active line (bottom-left plot of Fig. 6). Depending on which of the three key parameters is varied, i.e. $R_S$, $C_H$ or $C_{Int}$, the resulting waveform outputted by the passive line changes significantly. An increase in spread

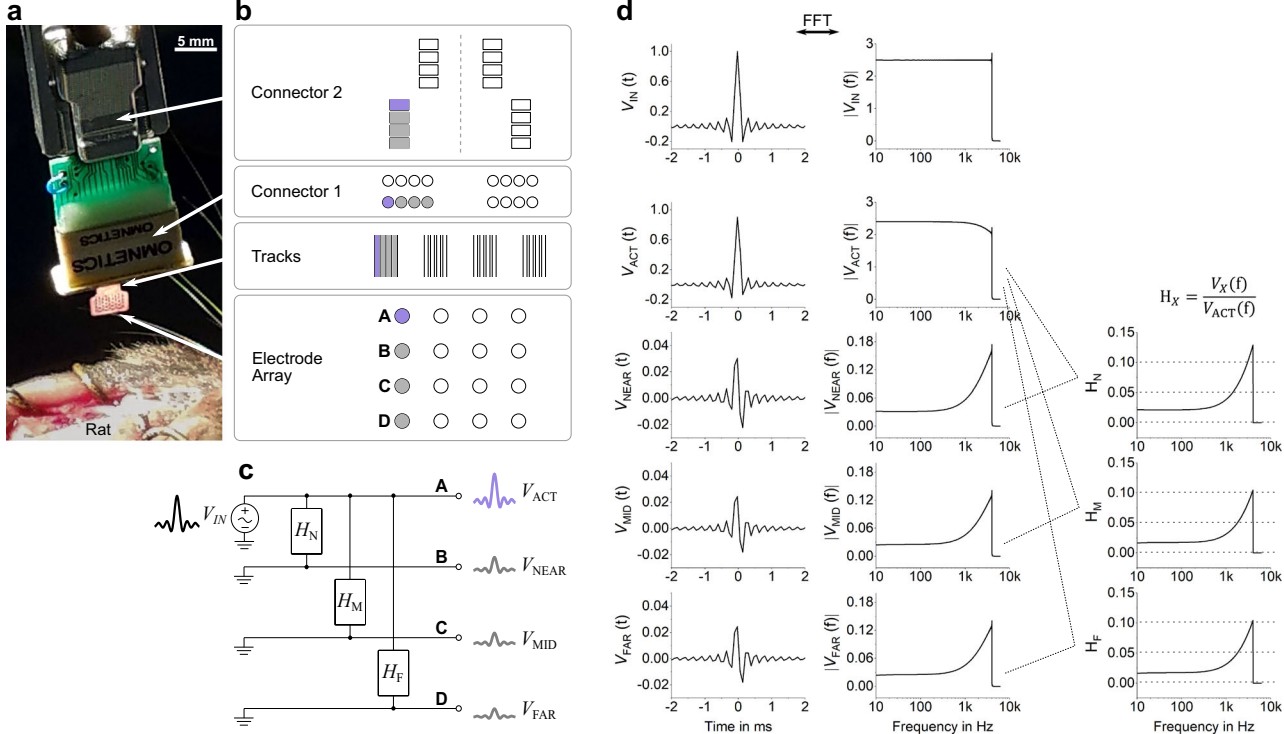

**Fig. 7 | The crosstalk back-correction algorithm. a** Picture of the recording setup, depicting the thin-film implant, the Omnetics-nano connector stage, the intermediate adapter board and the ZIF-Clip® headstage. **b** The recording channels are routed in sets of four throughout the whole recording chain. Taking a representative column of four electrodes, the top electrode A is defined as active (purple color), while the bottom three electrodes B, C and D are defined as passive (gray color). **c** Extension of the two-line transmission model from Fig. 4 to a four-line model, representing each set of four electrodes. $V_{IN}$ stands for the input signal, a sinc wave with 4 kHz bandwidth and 100 ms duration injected into the active line.

$V_{ACT}$, $V_{NEAR}$, $V_{MID}$ and $V_{FAR}$ stand for the output signals of the active line (ACT) and the three passive lines, named in accordance to their distance to ACT (NEAR, MID and FAR). The coupling between the active line and each of the three passive lines is simplified as three blocks with transfer function $H_N$ (between ACT and NEAR), $H_M$ (between ACT and MID) and $H_F$ (between ACT and FAR). **d** Transfer functions $H_N$, $H_M$ and $H_F$, computed as the ratio between the Fast Fourier Transform (FFT) of the output signal of the respective passive line ($V_{NEAR}$, $V_{MID}$ and $V_{FAR}$) and the FFT of the output signal of the active line ($V_{ACT}$).

resistance $R_S$ results in high-amplitude negative and positive peaks (peak-to-peak amplitude reaches 16% of the active signal), although slow oscillations are lost. A drop in insulation capacitance $C_{Int}$ also yields a sharp and high-amplitude negative peak but distinctly buffered positive peak. Very differently, a boost in double layer capacitance $C_H$ is not associated with particularly large peaks, but the slow components are highly preserved.

**Crosstalk back-correction algorithm**

The recording system used in this work is particularly suitable to study the influence of crosstalk on the recorded data because the routing layout is symmetrical along the whole chain. In fact, the four electrodes forming each column of the microelectrode grid (Fig. 3a) are neighbors throughout all stages of the recording system, from thin-film to Omnetics connector to ZIF-Clip® headstage, as shown in Fig. 7a, b. Therefore, crosstalk is boosted within these sets of four channels.

In principle, if the signal distortions discussed in Section 2.1 are indeed a consequence of crosstalk, then it should be possible to use the transfer function generated by the lumped-element model as a correction tool. This would allow extracting the true signals from the crosstalk-corrupted signals. In order to do so, the two-line model (Fig. 4) is extended to a four-line model, as shown in Fig. 7c, representing each set of four electrodes composing a column of the electrode grid. The top line is defined as the active (ACT) and the other three as passive, each of them named according to their distance to the active line: NEAR, MID and FAR.

A sinc wave with 4 kHz bandwidth and 100 ms duration is injected into the active line as input signal $V_{IN}$ (Fig. 7d). This waveform is

convenient for further processing in the frequency domain, since its Fourier transform is a rectangular pulse, thus having a constant magnitude in the whole frequency spectrum. The coupling between the active line and each of the passive lines can be described by individual transfer functions: $H_N$ for the near line, $H_M$ for the mid line and $H_F$ for the far line. Each transfer function is computed as the ratio between the FFT (fast Fourier transform) of the output signal of the respective passive line ($V_{NEAR}$, $V_{MID}$ and $V_{FAR}$) and the FFT of the output signal of the active line ($V_{ACT}$), as defined by the Eq. 1.1–3.

$$H_N = \frac{FFT(V_{NEAR})}{FFT(V_{ACT})} \tag{1.1}$$

$$H_N = \frac{FFT(V_{MID})}{FFT(V_{ACT})} \tag{1.2}$$

$$H_N = \frac{FFT(V_{FAR})}{FFT(V_{ACT})} \tag{1.3}$$

These three transfer functions describe the coupling between any two lines within any of the sets of four channels composing the recording system. The spectrum ranges up to 4 kHz, determined by the bandwidth of the input sinc wave (Fig. 7d).

The crosstalk-corrupted signals recorded by the four channels composing each column of the grid are hereby denominated A, B, C and D (Fig. 7b, c). Each channel is given by its true signal plus the true signal of each of the other three channels multiplied by the

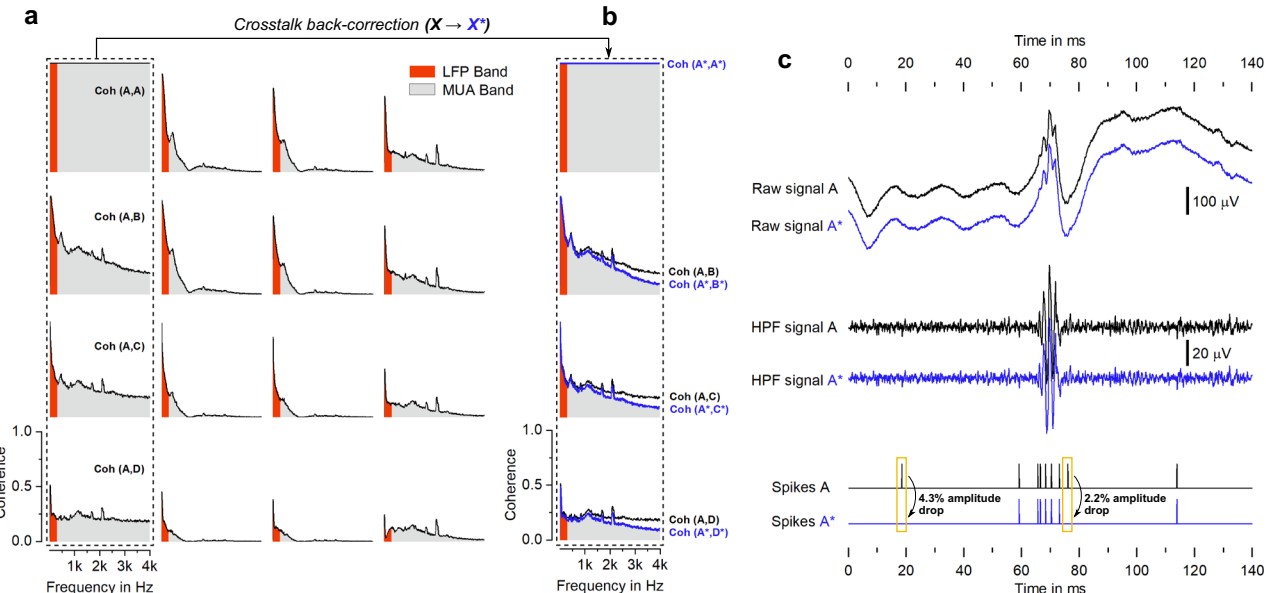

**Fig. 8 | Neural data reconstruction. a** Coherence map of the original crosstalk-corrupted signals, computed from 3 Hz to 4 kHz for the full recorded response of one implanted device and taking electrode A (top-left electrode of the grid) as reference. LFP and MUA bands are color-coded as red and gray, respectively. **b** Coherence before and after crosstalk back-correction, taking the first electrode column as a representative set (A, B, C and D). Signals after correction are symbolized by an asterisk (A*, B*, C* and D*). **c)** Raw signal, high-pass filtered (HPF) signal ($f_c$ = 300 Hz) and detected spikes referring to 140 ms s of induced somatosensory evoked activity recorded by electrode A before (black) and after (blue) correction.

appropriate transfer function, according to the system of four Eq. 2.1-4, where the true signals are marked by an asterisk.

$$
\begin{cases}
A = A^* + H_N \cdot B^* + H_M \cdot C^* + H_F \cdot D^* \\
B = H_N \cdot A^* + B^* + H_N \cdot C^* + H_M \cdot D^* \\
C = H_M \cdot A^* + H_N \cdot B^* + C^* + H_N \cdot D^* \\
D = H_F \cdot A^* + H_M \cdot B^* + H_N \cdot C^* + D^*
\end{cases}
\quad (2.1-4)
$$

Channels A and D are coupled to one near-distant line, one mid-distant line and one far-distant line, while channels B and C are coupled to two near-distant lines and one mid-distant line. Solving the system allows to extract the four true signals A*, B*, C* and D*.

**Validating algorithm with recorded ECoG data**

To validate the algorithm, ECoG recordings are analyzed and compared on the basis of signal coherence (Coh), a statistical quantity used to examine the relation between two data sets in terms of their spectral densities. Being specified in frequency, coherence allows to investigate how the similarity between channels changes before (X) and after (X*) correction in the two frequency bands of interest, i.e. LFP and MUA.

Taking the original signals, coherence plots are computed from 3 Hz to 4 kHz for all sixteen electrodes using electrode A as reference and displayed according to the electrode grid layout (Fig. 8a). In the LFP band (highlighted in red), coherence decays radially from the reference electrode. However, in the MUA band, coherence is clearly higher for the reference column, with a plateau at a coherence level of 0.3 above 3 kHz, thus indicating crosstalk contamination. The crosstalk back-correction is performed on the set of four electrodes composing the reference column (electrode 1, 5, 9 and 13) and the coherence computed on the corrected signals (Fig. 8b). Coherence levels drop significantly in the spike band, while remaining unaltered in the LFP band, as anticipated by the simulations predicting a significant coupling only above 1 kHz (Fig. 6).

It is furthermore relevant to validate whether the corrected signals retain their significance in terms of electrophysiological information. Figure 8c shows 140 ms s of evoked somatosensory activity

recorded by electrode A before and after correction. The raw and high-pass filtered ($f_c$ = 300 Hz) SEP waveforms after correction are indistinguishable from the originals. Ultimately, MUA is analyzed as spike activity, thus requiring spike detection, i.e. the task of identifying neural action potentials, or spikes, from the background noise. Although most spike events are common and coincident before and after correction, a few are, in fact, discarded in the process. Out of nine spikes originally represented in the considered time interval, two of them cease to be detected after correction due to a 4.3 % and 2.2 % drop in waveform amplitude. The present results suggest that, under the boundary conditions specific to this recording system, crosstalk contamination is strong enough to cause spurious detection of spikes in the active channel recording the closest from the signal source. Furthermore, it is important to note that, generally, the signals recorded by the passive channels, even though weaker in terms of information content, are proportionally more affected by crosstalk contamination precisely due to the smaller amplitude signals (Figs. S2 and S3 in the supplementary material).

## Discussion

### Understanding crosstalk and its impact on brain recordings

The routing layout of the multi-channel recording system employed in this work discernibly boosts crosstalk within sets of four channels. According to the electrode mapping, the four electrodes composing each column of the four-by-four grid are routed adjacently across all stages of the recording chain: polyimide substrate, Omnetics-nano connector stage and ZIF-Clip® headstage. In the end, electrical coupling between adjacently wired channels undoubtedly distorts the spectra of the respective recorded signals, resulting in highly increased correlation levels in the MUA band, even if the electrodes themselves are placed far away on the cortex. According to volume conduction, signal correlation should increase with proximity at the cortex level, but never with proximity at the routing level.

The crosstalk investigation conducted here calls attention to an important aspect. Crosstalk between active and passive lines occurs via the two paths that physically couple them to one another: through the insulation at the implant level and through the insulation at the

interconnects level. Nonetheless, the magnitude of this coupling depends on the complete transmission line, including the termination impedances, i.e. the tissue-electrode interface and amplifier's input impedance. In other words, the quality of the interface between recording electrode and signal source has a direct impact on the magnitude of coupling: high-impedance nodes experience more crosstalk than low-impedance nodes. This point stresses the fact that the transfer function is defined by the superposition of the different blocks, and its analysis must take into consideration the equivalent circuit in its entirety as a comprehensive and closed system, rather than taking each block as an independent, open subsystem. Here we study the effect of each block by looking at how its variation affects the baseline transfer function computed for a given set of starting parameters.

Indeed, the termination impedance on the tissue side is a crucial determinant of crosstalk, including both the spread resistance and the electrochemical interface between electrolyte and electrode. In the context of epicortical and, in particular, of epidural electrodes, it is useful to evaluate the spread resistance ($R_S$) separately from the resistance of the electrolyte in direct vicinity to the electrode ($R_E$). The dura mater is a thick insulating layer with a resistivity 25 times superior to that of cerebrospinal fluid[20], hence increasing the termination impedance. According to Newman et al., the resistance to flow of current to a disk electrode can be estimated as $R = \rho/(4r)$ (in $\Omega$), where $\rho$ stands for the medium resistivity (in $\Omega \cdot$m) and $r$ for the radius of the disk (in m)[21]. Given that the resistivity of the dura mater has a value of 1–5 k$\Omega \cdot$cm[20], the spread resistance for an epidural electrode disk with a radius of 50 µm is calculated to have a minimum value of 50 k$\Omega$, which matches the experimental value of 45 k$\Omega$, one order of magnitude larger than $R_E$ (Table 1). It is thus concluded that, in epidural applications, the meninges play a major role on the termination impedance of the transmission line and, consequently, on crosstalk influence.

The LFP band is found to be almost immune to crosstalk contamination, which is consistent with the fact that, under the boundary conditions dictated by our recording system, coupling levels are negligible in this frequency band, becoming significant, i.e. less than -40 dB attenuation, only above 1 kHz (Fig. 6, dashed baseline curve). Nonetheless, this holds true because the EIS data used to model the tissue-electrode interface is acquired on implantation day, before the foreign body response steps in. As edema and glial encapsulation, from the biotic side, and delamination and degradation of thin-films, from the abiotic side, come into play, the parameters of the model start deviating from the baseline values (increase in spread resistance $R_S$ and decrease in double layer capacitance $C_H$). Such parameter deviations lead to shifts in the transfer function, ultimately resulting in a drastic increase in crosstalk, inclusive in the LFP band. Also the insulation impedance at the implant level, in our case defined by the resistive and dielectric properties of polyimide, is highly susceptible to the development of the implant over time. As water from the surrounding tissue is absorbed into the polymer, its dielectric properties change, leading to an increase in admittance as shunt path from insulation to tissue and as coupling path between tracks. The rate of absorption varies substantially depending on the material, even within the same family of polymers. In our particular case, however, variations in the capacitance at the implant insulation level are not the most determinant to the transfer function.

Besides spread resistance and Helmholtz capacitance, insulation impedance at the interconnection level is, in this particular case study, the third main determinant of crosstalk. The first two depend on complex brain-implant interaction and, admittedly, their optimization remains rather challenging. Shortening the interconnection stage by implementing on-site amplification and digitization is an effective and straightforward approach to bring crosstalk levels down. Besides, it has the add-on effect of minimizing the path length

subjected to further sources of interference, e.g. power line interference[22,23].

## Reconstructing crosstalk-corrupted neural data

The generalized two-line model used to characterize any recording system is extended to a four-line model simulating our particular setup, where sets of four adjacently routed channels suffer from increased crosstalk contamination. According to the clearance between the active line and the three passive lines (increasingly distant from the active line), three transfer functions are computed, describing the coupling between two near-distant lines, two mid-distant lines and two far-distant lines (Fig. 7d). From this simplified model, an algorithm for crosstalk back-correction is developed. The signal recorded by each line is assumed as a superposition of its true signal and the true signal of each of the other three lines multiplied by the respective transfer function. By solving the system of four equations, the true signals can be extracted and compared to the original signals contaminated by crosstalk. The present algorithm serves as a method to verify the hypothesis that electrical coupling is strong enough to modulate the signal spectrum within levels relevant to the field. However, it is not intended as a method to surpass crosstalk contamination a posteriori. On the contrary, it should raise awareness to mitigate it at its source.

Coherence levels between signals recorded by adjacently wired channels are falsely boosted in respect to all other channels. Yet, after performing the crosstalk back-correction algorithm on these signals, coherence effectively drops in the MUA band, while remaining unchanged in the LFP band. Furthermore, comparing recorded and reconstructed raw data shows that coupled signals are strong enough to modulate waveform amplitudes by as much as 4.3%. This could have serious consequences on signal analysis and, ultimately, on the extraction of electrophysiologically relevant information, as the shift in amplitudes is high enough to sway spike waveforms beyond the detection threshold. Any further conclusions on the impact of the crosstalk contamination on validity of spike detection and information extraction would require dedicated studies with defined criteria beyond the scope of this work.

Strong coupled signals not only distort the original waveforms due to superposition of time series, but they also create an erroneous link between the passive line and the true brain activity picked up by the active line, leading to misinterpretation of data and loss of spatial discrimination between recording channels. For example, let us take an implant on implantation day. The tissue-electrode interface is pristine and all electrodes are recording at a high signal-to-noise ratio. The number of spuriously detected spikes due to crosstalk are likely to be insignificant in proportion to the true activity being picked up. Now, let us move to advanced chronic stages, where a percentage of electrodes is failing due to glial encapsulation and thin-film delamination. Because of the extremely high terminal impedance on the tissue side, these lines behave as strong passive lines. If an adjacent line happens to interface a still well-transducing electrode, then the passive line, which should be outputting zero, is now recording a muffled version of the active neighbor. Being that, in chronic stages, the tendency is for signal-to-noise ratio to drop and variability between electrodes to increase, coupled signals might be easily mistaken for valid residual information. Therefore, cross-checking the recordings against the routing layout becomes imperative, as it can help discern and discard crosstalk contamination.

## Guidelines to ensure high quality, crosstalk-free recordings

This study highlights the fact that the spectrum of the data outputted by a recording system, which should ideally reflect the signal source alone, is in fact conditioned by all building blocks making up the system, from tissue-electrode interface to amplifier. When designing a recording system, there is a broad consensus about what one should strive for: in simple terms, minimize impedance at the tissue-electrode

interface, maximize conductance of interconnects and maximize insulation impedance. The elements defining the interconnects and amplifier's input impedance remain typically unchanged throughout the recording period, while elements defining the neural implant are dynamic and evolve along the course of the implantation, with chronic settings being the most critical. This in turn means that the transfer function of the recording system is constantly evolving and will be unique to each setting.

Pinpointing design rules for such complex systems is difficult, because they are in practice a compromise between many variables. For example, electrode miniaturization has the goal of increasing spatial selectivity and extending the recorded frequency bandwidth, but it comes at the expense of higher tissue-electrode impedance. This justifies all the research going into developing superior electrode materials and coatings that ultimately lower the electrode impedance per unit area[24]. It is therefore impractical to put a threshold on what is an acceptable tissue-electrode impedance, as it heavily depends on the application and type of data being acquired. Similarly, a trade-off can be identified for the insulation impedance of both implant and interconnects. On one hand increasing the number of recording channels and density, on the other hand using soft, flexible and ultra-thin substrates to minimize the mechanical mismatch to the tissue, ultimately pushing the boundaries of an effective insulation between channels. It is also relevant to consider that all these building blocks have capacitive components, meaning they are frequency dependent. While crosstalk increases with frequency, electrode impedance on the other hand decreases with frequency due its dependency on capacitive displacement currents, which in turn reduces coupling. Importantly, impedance threshold values for single elements at specific frequencies fail to take into consideration how all blocks play out together when connected, and in dependency of the frequency being recorded. Plugs and connectors which have to be connected for recording sessions, for example, are quite often completely neglected. However, fluid and particles could strongly change contact as well as pin-to-pin impedance. Applied forces could lead to deterioration of the contact with increased resistance and decreased insulation properties due to adhesion failure of the encapsulant (epoxy or dental cement, for example). These long-term effects go far beyond the scope of this study but should also be kept in the mind when conducting chronic experiments with transcutaneous headstages.

Following a different approach, this work encourages users to consider crosstalk as a key specification defining the quality of a recording system, just as it is routinely done for active neural implants. For the purpose of this study, we have computed crosstalk from a lumped-element model, with all its building blocks being applicable to any other system. However, a more straightforward way to quantify crosstalk of a transmission line is to use dedicated equipment, such as a network analyzer. As a general rule, attenuations of <−40 dB become significant. On the data side, we emphasize the importance of validating baseline recordings before diving into further analysis. Computing signal correlation and, more specifically, spike cross-correlation and checking those against the routing layout is a good method to exclude crosstalk contamination, as they are found to be indicators discerning crosstalk influence from volume conduction. As a reminder, here we use a crosstalk-back correction algorithm to validate our hypothesis, but not to encourage its use as a strategy to restore low-quality, crosstalk-contaminated data.

## Concluding summary

Considering the increasing demand for miniaturization and higher spatial selectivity in neural recordings, it is essential to ensure that recorded signals remain genuine to the local activity underlying each single electrode. Coupling and blending of signals emerging from different cortical spots must be avoided at all costs, as it results in reduced spatial discrimination, thus defying the purpose of higher

integration densities. Our study demonstrates that epicortical MUA recordings from anesthetized rat brains, obtained with passive thin-film implants and a commercial readout system, are compromised by crosstalk occurring along the brain-to-amplifier chain. Indeed, understanding the direct impact of miniaturization on the quality of recorded neural data is fundamental. One thing is to have a functional implant and one other entirely is to have an optimized technology that transfers signals from brain to computer with minimal information loss in between.

This work emphasizes the importance of electrically characterizing a recording system before signal acquisition. Not only the neural implant itself should be considered, but all blocks building the chain up to the amplifier, as limiting coupling paths may be found at higher levels in the recording chain. The lumped-element model proposed here serves as a generalized and valuable toolbox that can be parametrized for any readout system. Parameters can be tuned to test particular scenarios, such as the biotic encapsulation of an electrode or breakage of the insulation layer, making crosstalk assessment more pragmatic. Here, for the first time, modeling and signal analysis are bridged together via a crosstalk back-correction algorithm developed to reconstruct real neural data in an effort to show how parametrization of the system can have a true effect on the outputted data.

In summary, the value of this study resides on the following points: (1) model parametrization is done in a step-by-step and easily transferable manner, where all lumped elements are backed-up by experimental impedance spectroscopy data; (2) signal correlation and, more specifically, spike cross-correlation are found to be indicators that differentiate crosstalk influence from volume conduction; (3) an unprecedented crosstalk back-correction algorithm is developed as a method to validate our crosstalk contamination hypothesis. These findings can contribute to changing the approach towards optimization and standardization of neural acquisition systems, and hence maximize the recording potential of modern neural implants. Indeed, mitigating crosstalk supports accurate signal transfer from brain to computer, helping ensure acquisition of ground truth data to decipher functions of the brain.

## Methods
### Thin-film device fabrication
The thin-film devices were fabricated using standard microlithography techniques in the clean-room facility (ISO 5, according to ISO 14644−1) at the Department of Microsystem Engineering (IMTEK) of the University of Freiburg. The fabrication process started with spin-coating of polyimide (U-Varnish-S, UBE Industries, Ltd, Japan) on a bare silicon wafer at a speed of 4500 rpm, providing a film thickness of 4 μm. Afterwards, curing was done at 450 °C. The high precision image-reversal photoresist AZ 5214 E (Micro-Chemicals GmbH, Germany) was used as sacrificial layer for the patterning of the metallization layer, which included electrodes, tracks and pads. An oxygen plasma surface activation was performed to promote adhesion (80 W, Plasma System 300-E, PVA TePla, Germany) prior to the evaporation of a 300 nm-thick platinum layer (Univex 500 Electron-Beam Evaporator, Leybold GmbH, Germany). The metal was structured in a lift-off step consisting of three acetone baths. After another oxygen plasma activation, a second polyimide layer of 4 μm was spun and cured. For patterning of the openings and outlines, the thick positive resist AZ 9260 (Micro-Chemicals GmbH, Germany) was used as masking layer. The structuring was done by reactive-ion-etching (STS Multiplex ICP, SPTS Technologies, United Kingdom) in a two-step process, first at 200 W and then at 100 W. The remaining resist was then stripped in acetone. Details on the design of the probe are provided in Fig. 9.

### Connector assembly
The thin-film device was mounted and assembled on a connector stage, as sketched in Fig. 9. Five layers of HTCC (high temperature

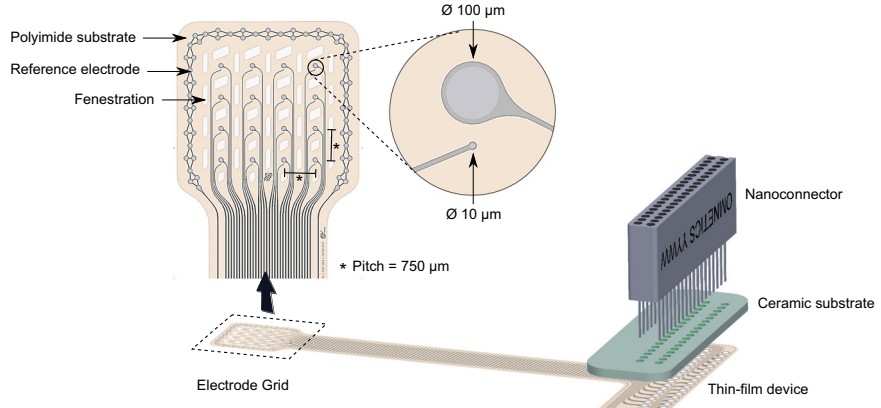

**Fig. 9 | Implant design and connector assembly[6].** The polyimide-based device is 8 μm-thick and features a platinum metallization layer. The recording area comprises a four-by-four array (vertical and horizontal pitch of 750 μm) and each position of the array includes one large electrode (100 μm in diameter) and one small electrode (10 μm in diameter). For this study, only the large electrodes were used. Furthermore, an on-site reference electrode is included for bipolar recording configuration (not used for this study). Fenestrations on the polyimide substrate help with the drainage of fluids and promote the growth of fibrotic tissue through the holes and around the device, rather than in the space between tissue and electrodes. For mechanical stability, the thin-film interfaces the male Omnetics connector via an intermediate ceramic board.

co-fired ceramic) precursor tape (44000, ESL ElectroScience, Pennsylvania, USA) were cut and pressed at 27 MPa, resulting in a single layer roughly 1 mm-thick. The outlines and holes through which the connector pins were meant to be inserted were laser patterned on the tape (DPL Genesis Marker, cab Produkttechnik GmbH, Germany). After an organic burnout at 550 °C and sintering at 1500 °C, silver-palladium paste was used to metalize the laser-structured holes. The paste was pressed into the holes from the top side of the ceramic and vacuumed from the bottom side, resulting in metallization of the inner walls of the holes. The paste was fired at 850 °C, followed by a grinding step on each side of the ceramic to remove the paste excess. The through-hole connector (A79022-001, Omnetics Connector Corporation, Minneapolis, USA) was then inserted into the ceramic and soldered. The solder bumps were grinded until flat. For mechanical stability and insulation, epoxy (EPO-TEK® 353ND-T, Epoxy Technology Inc., Massachusetts, USA) was spread around the edge between connector and ceramic. Lastly, to mount the thin-film on the connector stage, the pads of the thin-film were aligned to the flat contacts on the ceramic and subsequently soldered using low-temperature soldering paste. The solder bumps were protected and insulated with fast-curing two-component epoxy (UHU® Plus Schnellfest, UHU GmbH, Germany).

## Animals and implantation procedure

The experimental plan was designed in compliance with the guidelines established by the European Communities Council (Directive 2010/63/EU, Italian Legislative Decree n. 26, 4/3/2014) and the protocol was approved by the Ethics Committee for animal research of the University of Ferrara and by the Italian Ministry of Health (authorization n. 989/2020-PR). Four adult Long Evans rats with 400–500 g were anesthetized with a mixture of Zoletil (Virbac, France; 30 mg·kg-1) and Xylazine (Bayer, Germany; 5 mg·kg-1) administered intraperitoneally. For the duration of the whole procedure, the depth of anesthesia was monitored by testing the absence of hind limb withdrawal reflex and maintained by additional intramuscular doses of anesthetic. The anesthetized animal was placed in a stereotaxic apparatus (David Kopf Instruments, California, USA) equipped with ear bars (Model 957 for small animals) and $a \approx 2$ cm-long incision was done along the midline of the cranium. The underlying muscle and connective tissue were retracted to expose the skull and a craniotomy $\approx 6 \times 6$ mm$^2$ was made in the parietal bone to expose the somatosensory cortex (identified by vascular landmarks and stereotaxic coordinates). Sterile saline solution was dispensed while drilling to avoid any local heating and to keep the bone surface clean. The thin-film was placed over the dura mater in the primary somatosensory barrel field (S1BF) and covered with Kwik-Sil silicone polymer (World Precision Instruments, Florida, USA) for protection. Five stainless steel bone screws were inserted in the skull, with one serving as ground electrode. The Omnetics connector was fixed to a custom-made chamber fabricated with a 3D printer (Voxel8 Inc., Massachusetts, USA), which was cemented to the skull and screws using dental acrylic (Jet Repair Acrylic, Lang Dental Manufacturing, Illinois, USA). Finally, the skin was sutured around the implant.

## Rat recordings and sensory stimulation

Neural recordings were collected on implantation day from four anesthetized and head-fixed rats using a Tucker Davis Technologies multi-channel recording system 3 (Tucker Davis Technologies Inc., Florida, USA) including the ZIF-Clip® headstage with unity gain, the PZ2-256 battery-powered preamplifier and the RZ2 real-time processor. Data were digitized at a sample rate of 12207 samples per second at 18-bit resolution and transmitted from the PZ2 preamplifier to the RZ2 processor by fast fiber optic connection. Using unipolar recording configuration, reference and ground pins of the headstage were tied together and connected to a skull screw. A Faraday cage was used to reduce electromagnetic interference.

To elicit the neural response of the barrel cortex, a vibrating system was used to produce deflection of the whiskers along the horizontal plane. Whiskers contralateral to the craniotomy were shortened and fixed to a Velcro strip attached to a rod. For multi-whisker stimulation, all whiskers were fixed to the strip, while, in the case of single-whisker stimulation, only one whisker was attached. The rod was moved by a shaker (Type 4810 Mini Shaker, Bruel & Kjaer, Denmark) controlled by a National Instruments board (Texas, USA). The deflection stimulus consisted of a sine waveform with a duration of 12 ms and an amplitude matching a deflection of 500 μm and was repeated 60 times with 4 s pause between repetitions.

## Rat data analysis

Neural signal analysis was performed using NeuroExplorer® software (v. 5.211, Nex Technologies, USA) and MATLAB workspace (v. R2019b, MathWorks, Massachusetts, USA). Signals were time-locked to the start of whisker stimulation ($n = 60$). Data were band-pass filtered between 3 Hz and 300 Hz for LFP analysis and between 300 Hz and 6 kHz for MUA analysis (Butterworth filter of fourth-order). To verify the consistency of the multi-unit activity recorded from the surface of the

cortex after the peripheral stimulation, spike detection was performed in accordance with parameters from literature (analysis *Detect Spikes* in NeuroExplorer)[9]. A -3 standard deviation threshold was set and no refractory period was assumed. The detected spike times were visualized by computing peri-event time histograms (analysis *Perievent Histograms* in NeuroExplorer). The histograms showed the counts/bin at time $t_0 + t$ with $t_0$ as the start of whisker stimulation (trigger) and $t$ as the defined bin length (1 ms). Cross-correlograms of spike activity were also computed (analysis *Crosscorrelograms* in NeuroExplorer), giving the conditional probability of a spike at time $t_0 + t$ with $t_0$ as the detected spike events of one specific electrode (reference) and $t$ as the defined bin length (1 ms). To avoid saturation of the maximum global value at $t_0$ for the reference electrode (probability of 1), reference spikes were not counted when calculating cross-correlogram versus itself. Signal correlation, defined as a normalized measure of the similarity of two time-dependent signals, was calculated as the Pearson correlation coefficient (function *corrcoef* in MATLAB). The coherence maps, understood as a measure of the degree of relationship between two time series as a function of frequency, were calculated from 3 Hz to 4 kHz (function *mscohere* in MATLAB), considering the whole recording session (256 frequency values, Hamming windowing function and window overlap of 50%).

## Impedance spectroscopy

The spread resistance between signal source (within brain cortex) and epidural recording electrode ($Z_1$) was estimated as the access resistance extracted from an impedance measurement performed between an intracortical needle inserted 0.5 mm deep into the cortex (working electrode) and an epidural electrode (counter electrode). The electrode impedance ($Z_2$) was measured in vivo on implantation day. A two-electrode setup was implemented, using a stainless-steel screw fixed on the animal's skull as a counter electrode. The potentiostat/galvanostat Reference 600 from Gamry Instruments Inc (Pennsylvania, USA) was used for both measurements. In potentiostatic mode, a sine wave with root-mean-square amplitude of 100 mV was imposed to the open circuit potential (OCP) and the frequency was swept from 1–100 kHz. Fitting was done using the software Gamry Echem Analyst (v. 6,32, Gamry Instruments Inc, Pennsylvania, USA).

The shunt impedance to the electrolyte ($Z_3$) was measured in vitro in a three-electrode setup using the potentiostat Autolab PGSTAT (Metrohm Autolab BV, Netherlands). An insulated (i.e. non-functional) electrode submersed is phosphate-buffered saline (PBS) 0.01 M solution (pH 7.4, Sigma-Aldrich, Missouri, USA) was used as working electrode, an Ag/AgCl (KCl 3 mol·l⁻¹) electrode as reference and a platinum wire as counter. A 100 mV sine wave was applied at open circuit potential (OCP) and the frequency swept from 1 Hz to 100 kHz. Fitting was done using the software Nova (Metrohm Autolab BV, Netherlands).

The insulation impedance between channels at the implant level ($Z_4$) was measured using a precision impedance analyzer (Agilent 4294 A, Keysight Technologies, California, USA). The analyzer has four measuring ports (4-point measurements): two leads connect to the current source (high and low force), while two connect to the voltmeter (high and low sense). By shorting the high force with the high sense lead and the low force with the low sense lead, the setup effectively functions as a 2-point measurement (two lead pairs). Each lead pair was connected to one of two neighboring tracks on the connector end. The frequency was swept from 40 Hz (minimum start frequency) to 1 MHz and the oscillation amplitude was set to 100 mV. A Faraday cage was used to reduce electromagnetic interference. The insulation impedance was quantified in terms of magnitude and phase, as well as an estimated capacitance and parallel resistance.

The insulation impedance between channels at the interconnects level ($Z_5$) was measured in a two-electrode configuration using the potentiostat/galvanostat Reference 600 from Gamry Instruments Inc

(Pennsylvania, USA). Fitting was done using the software Gamry Echem Analyst (v. 6,32, Gamry Instruments Inc, Pennsylvania, USA).

All setups used for impedance spectroscopy characterization are summarized as supplemental information (Fig. S4). A fitting analysis providing fitting errors and goodness of fit is summarized as supplemental information (Table S1).

## Lumped-element modeling and simulation

The transmission line was modeled in Simulink (v. R2019b, MathWorks, Massachusetts, USA) with the powergui set to continuous. The transfer function was computed using the linear analysis tool and saved in the MATLAB workspace (v. R2019b, MathWorks, Massachusetts, USA). For external input signals (spike waveform or sinc waveform), the signal was imported from the MATLAB workspace with the sample time set to 0 (default). The output signals were saved to the workspace with the sample time set to the 1/12207 s (same as recorded neural data). The algorithm for crosstalk back-correction of neural data was created in MATLAB.

## Reporting summary

Further information on research design is available in the Nature Portfolio Reporting Summary linked to this article.

## Data availability

All data supporting the findings of this study are available within the article and its supplementary files. Any additional requests for information can be directed to, and will be fulfilled by, the corresponding authors. Raw data of recordings and impedance spectroscopies do not contain information which could be used in a general context since the crosstalk is dependent on every particular device. Source data are provided with this paper.

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

## Acknowledgements

This work was supported by the BrainLinks-BrainTools program, which was funded by the German Research Foundation (DFG, Grant No. EXC 1086) and is currently funded by the Federal Ministry of Economics, Science and Arts of Baden-Württemberg within the sustainability program for projects of the excellence initiative II. This work was also partially supported by PNRR-POC-2022-12376380 to L.F. This project has received funding from the European Union's Horizon-EIC-2022-Pathfinder Open program under grant agreement No. 101099366 (BioFINE). Funding goes to M.A. and T.S. The authors would like to thank Dr.-Ing. Daniel De Dorigo from the Laboratory for Microelectronics at IMTEK, University of Freiburg for the support developing the two-line electrical model.

## Author contributions

M.F.P.C. wrote the paper; M.F.P.C., M.V., T.S. and L.F. conceptualized the work; M.F.P.C., E.Z. and M.V., designed the experiments; M.F.P.C., E.Z. and I.G.V. acquired, analyzed and interpreted data; M.F.P.C., E.Z., A.P., E.D. and M.DL. developed the algorithm; M.F.P.C. and A.P. implemented the algorithm; E.Z., M.V., A.P., I.G.V., E.D., M.DL., M.A., L.F. and T.S. revised the paper; M.A., L.F. and T.S. provided scientific advice; L.F. and T.S. funded the work.

## Funding

## Competing interests

The authors declare no competing interests.
