## [Transparent Peer Review file · Nature Communications]

Bridging circuit modeling and signal analysis to understand the risk of crosstalk contamination in brain recordings

Corresponding Author: Dr Maria Porto Cruz

Version 0:

Reviewer comments:

Reviewer #1

(Remarks to the Author)

Summary:

In this study, the authors tackle the challenge of crosstalk contamination in brain recordings from anaesthetized rat brains, an issue magnified using high-density microelectrode arrays in modern neurotechnologies. They identify crosstalk through signal coherence maps linked to the routing layout and introduce a back-correction algorithm to simulate a zero-crosstalk scenario, effectively demonstrating a decrease in signal coherence among closely routed channels upon correction. This validation against the routing layout is emphasized as crucial for data quality control, aiming to ensure the accuracy of brain activity data captured from complex recording systems. Their work addresses a timely question in neural recordings: maintaining the integrity of data amidst the complexities introduced by advanced recording technologies.

Comments:

1. The overall scope of the paper is highly technical. On one hand, I think the results are important because they systemically explore crosstalk contamination starting by discerning signals and understanding their correlation by modelling the various elements of the equivalent circuits in the signal chain. On the other hand, the orthogonality of the analysis based on different lumped elements, by simulating crosstalk and using the element model to propose a post-hoc correction algorithm, all validated against ECoG data makes the study highly technical.
2. Looking at the work critically, the algorithm operates under the hypothesis that it can accurately solve for the transfer functions between active and passive lines. This is based on the extension of the two-line transmission model to four-line, which is specific to the design of the routing in this implant. However, it would be beneficial to comment on how any given design of the implant can be extrapolated to obtain transfer functions for correction. Even better, it would help the audience to follow general design rules, which the authors should take the opportunity to propose to enable easy estimation of such correlations. Rules such as routing certain electrodes on the implant in a particular way and maintaining the same order of routing on connectors, PCBs, and so on.
3. Another clarification needed is on the validation of electrophysiological information in the MUA band, on the detection of spiking activity. The authors demonstrated that the number of spikes detected drops after correction due to a drop in amplitude. The authors hint that this is due to spurious detection of spikes from the original signal with crosstalk, but would it be possible that it could be an overcompensation from the algorithm potentially causing loss of relevant data? Maybe including additional criteria such as the refractory period in spike detection could accurately pinpoint to the nature of spikes? If the authors are indeed confident that these are glitches due to contamination, are there any suggestions to curb this?
4. Do the authors have any underlying explanation for the dominant crosstalk contamination above 300 Hz and the increase in signal coupling with increasing frequency in the MUA band compared to the LFP band? It would be beneficial to address this in the discussion section.
5. The study is performed on an acute implantation with all the impedance spectroscopies calculated around the same period. Could the authors please clarify the effect of ageing in the case of a chronic implant in terms of the change of dielectric properties of the insulation affecting both Z3 and Z4? In the simulation, the authors observe minimal variations in

signal coupling with increase/decrease to C_{sh} and C_{imp} but would R_{sh} or R_{imp} play any role?

6. Figure. 1 could have the 'r' represented using a dotted line.

7. Figure. 7 has been mistakenly labelled as Figure. 6 (second time).

8. Under methods – impedance spectroscopy – for Z_4 : the description reads 'One lead pair was connected to the left end of one track and the other to the right end of the second track' but in the supplementary figure. S1 shows both C and W extracted across the pins of the connector. Could the authors please clarify what is the left end and the right end per the figure?

Reviewer #2

(Remarks to the Author)

In the present manuscript the authors highlight the importance of crosstalk for brain-recordings, discuss its origin, and develop a de-embedding method to suppress the undesired influence. The paper is timely written and the topic of crosstalk for such recordings is in general very interesting. However, I cannot recommend the manuscript for publication in Nature Communications as I do not think that the paper will have a sufficient impact in the field. Furthermore, the manuscript lacks clarity in the description of several aspects.

The main learning is that R_S and C_H dominate the cross-coupling. However, they are not coupling between the passive and active lines. Only the Z_n , $n>2$ components do that. Of course the authors also discuss the role of Z_5 . However, this coupling is a 2nd order effect and I think that it can be minimized by the choice of proper connections. The coupling through R_S and C_H is no undesired coupling. Instead, it is essential to create a map of brain activity.

I list a couple of other points below (not ranked by importance)

1) Tests in Figure 2: Why do only electrodes 5,9, and 13 show strong correlation behavior and not e.g., 2, 3 and 6. How are the electrodes arranged on the micro-electrode disc?

2) What is the thickness of the 2nd PI layer, which is used to cover the Pt electrodes? Also $4\mu\text{m}$?

3) Pt electrodes are commonly used for brain recordings. However, there are better electrode materials available and the authors should discuss the relevance of crosstalk for such electrodes. Furthermore, the authors do not comment on the performance of their Pt electrodes and how their transition impedance compares to the state-of-the-art. Crosstalk of electrodes is most relevant when the impedance of R_S and C_H are high (bad contacts). But they become negligible with good electrodes.

4) the authors should provide fitting errors for their impedance analysis. In particular, I doubt that they can fit R_{IMP} and R_{int} in a meaningful way.

5) As the authors discuss, variations in R_S , C_H and C_{int} have the largest influence on the impedance recording in the frequency range $<1\text{kHz}$. In particular, C_H can easily vary from experiment to experiment depending e.g., on the ion concentration in the biological environment and details of the electrodes. Due to the measured value of C_{int} , it also has a large influence on the reading. Although a 10x increase or decrease of C_{int} is theoretically possible, I doubt that this is going to happen in experiment. So, do you think that a 10fold variation of C_{int} is realistic in experiment? I would doubt that.

6) Furthermore, combining the finding that only the MUA range is affected by x-talk and not the LFP (Figure 2), I would conclude that R_S and C_H are most important for coupling and not C_{int} (at it would also influence the LFP according to the spectra in Figure 5). So, following the impedance spectroscopy analysis, the coupling components Z_n , $n>2$ can be disregarded as they are typically at least 10x higher in impedance than Z_n , $n<3$. Which means that there is not coupling between V_{ACT} and V_{PAS} ?

Reviewer #3

(Remarks to the Author)

What are the noteworthy results?

The authors study the effects of crosstalk in epicortical recordings. The authors present a model to estimate the crosstalk in adjacent electrodes and where the adjacent electrodes are compromised by crosstalk along the brain-to-amplifier chain. Based on the model an algorithm is presented to extract the true signals from the crosstalk-corrupted recordings. The model and algorithm were evaluated in-vivo. The work is interesting and emphasizes the importance of crosstalk consideration. Reducing the cross-talk is obviously an important endeavour. This topic has, in fact, been previously investigated in several studies (see next section). However, this study demonstrates the agreement of theory with in vivo recordings.

Will the work be of significance to the field and related fields? How does it compare to the established literature? If the work is not original, please provide relevant references.

Crosstalk in neural recordings has been investigated in other publications. A simplified scheme which models the crosstalk from a brain-to-amplifier chain was already proposed by Najafi et al. (1990) and further developed in Du et al. (2009) and, in Seidl et al. (2012). Crosstalk in neural electrodes has been investigated without an electric circuit model by Qiang et al. (2021) (cited by authors) and Naughton et al. (2022). I suggest that the authors refer back to some of these relevant works and, importantly the original article where the model was first presented.

Despite the above, the paper represents a novelty by extending/using an electrical equivalent model to estimate the crosstalk and a proposed algorithm based on the model to extract the non-contaminated signals. The usefulness of this algorithm is somewhat limited; it is not suitable as a method for subsequently overcoming crosstalk contamination. It is here employed as a method to verify the hypothesis that the electrical coupling is strong enough to modulate the signal spectrum, and to understand to what extent the contamination by crosstalk may affect the extracted neural information.

Overall, in the view of the reviewer, the hypothesis is only conditionally new (also the paper builds on the previously published work by the authors: M. F. Porto Cruz et al., "Can Crosstalk Compromise the Recording of High-Frequency Neural Signals?," 2019 9th International IEEE/EMBS Conference on Neural Engineering (NER), San Francisco, CA, USA, 2019, pp. 924-927, doi: 10.1109/NER.2019.8717009.), but the detailed investigation of the model, brain-amplifier chain and the associated frequency analysis are of great importance.

Relevant references

K. Najafi, J. Ji, and K. Wise, "Scaling limitations of silicon multichannel recording probes," *IEEE Trans. Biomed. Eng.*, vol. 37, no. 1, pp. 1–11, Jan. 1990.

J. Du, I. H. Riedel-Kruse, J. C. Nawroth, M. L. Roukes, G. Laurent, and S. C. Masmanidis, "High-resolution three-dimensional extracellular recording of neuronal activity with microfabricated electrode arrays," *J. Neurophysiol.*, vol. 101, no. 3, pp. 1671–1678, Mar. 2009.

Seidl, K., Schwaerzle, M., Ulbert, I., Neves, H. P., Paul, O., and Ruther, P. (2012). CMOS-based high-density silicon microprobe arrays for electronic depth control in intracortical neural recording-characterization and application. *J. Microelectromech. Syst.* 21, 1426–1435

Naughton J R, Varela J A, Connolly T J, Shepard S, Dodge T E, Kempa K, Burns M J, Christianson J P and Naughton M J 2022 Suppression of crosstalk in multielectrode arrays with local shielding *Front. Nanotechnol.* 4 1–11

Does the work support the conclusions and claims, or is additional evidence needed?

Overall, the authors present a thoroughly conducted series of experiments that confirm their claims. No additional evidence needed.

However, it would be useful to see the level of crosstalk contamination after each stage in the simulations. For example, how much does the spread resistance really contribute?

Are there any flaws in the data analysis, interpretation and conclusions? Do these prohibit publication or require revision?

Data analysis seems sound and justified. However, there are still some open questions that require clarifications, as below:

- o In figure 6, and section 2.5. electrode A (reference) is used as an example. Electrode A is indicated as the active electrode. A is also used as an example in Figure 6 c) for the correction algorithm. This indicates there is crosstalk from the electrode (B,C,D) (or all) to A. All other electrodes but A would be a more suitable electrode to analyse.
- o Also waveform cross-correlations, as in section 2.1 (Figure 2.), should be reported. The same analysis for channels B,C, and D should be reported in the supplementary.
- o Additionally, how significant is the reduction in the coherence maps with the algorithm? Please include some statistics on that.

Is the methodology sound? Does the work meet the expected standards in your field?

The authors present a valid methodology from initial experiments from where a hypothesis is derived. Furthermore, they support their hypothesis with a model and a model validation with in-vivo experiments.

In general, including in the experiment an isolated electrode (no opening) as a neighbour, would have given a more accurate representation of cross-talk coming from adjacent active lines/electrodes/interconnects.

Is there enough detail provided in the methods for the work to be reproduced?

Some fundamental theoretical background crosstalk principles and analysis is missing. E.g., it is known and expected that coupling increases with frequency. Could the authors please add this frame in the paper so that the reader can better appreciate and by extension replicate such analysis for a different setup?

The algorithms for coherence maps and correlograms of spikes with the Matlab code and data should be available.

Other feedback/comments/typos

- Statement on page 19., section 3.1 Paragraph 1: "However, it is not intended as a method to surpass crosstalk contamination a posteriori. On the contrary, it should raise awareness to mitigate it at its source." Can you elaborate on that, what are the limitations?
- Statement on page 20, the authors claim to crosscheck crosstalk against the routing layout: "Therefore, cross-checking the recordings against the routing layout becomes imperative, as it can help discern and discard crosstalk contamination." How do the authors propose to address this quantitatively in a realistic scenario?
- Do electrodes A,B,C,D in figure 6 (by the way there are 2 figures numbered 6, please correct numbering) represent electrode 1,5,10,13 in figure 2, please clarify

- Ground electrodes and reference electrodes are common mistakes, and in this scenario, the role of a ground electrode and reference electrode should be explained. Especially since the term reference electrode is used interchangeably. As stated on page 24: " Furthermore, an on-site reference electrode is included for bipolar recording configuration (not used for this study)." Furthermore, on page 25: " Using unipolar recording configuration, reference and ground pins of the headstage were tied together and connected to a skull screw." and on page 16: "Taking the original signals, coherence plots are computed from 3 Hz to 4 kHz for all sixteen electrodes using electrode A as reference". The model, therefore, is connected to one ground, so the brain is grounded to the same ground as the amplifier and the actual reference electrode is connected to the ground.

It should be stated that the actual reference/ground electrode is not electrode 1 (A or reference) and that the actual reference/ground electrode placing has an impact on the recorded signal also related to crosstalk

- Page 4, paragraph 1: on filling or to fill
- Figure 1, c: Please increase waveforms and amplitudes, maybe include the uncertainties in the cross-correlations.
- Figure 6 occurs twice.
- Methods section 8.6: What are low force and low sense leads?

Reviewer #4

(Remarks to the Author)

Version 1:

Reviewer comments:

Reviewer #1

(Remarks to the Author)

I believe that the authors have addressed all comments from the previous review in a satisfactory way and the paper can be published.

Reviewer #2

(Remarks to the Author)

I thank the authors for the careful revision of the manuscript. They addressed all my comments in a satisfy way. However, I am still not very enthusiastic about the paper as I find the paper rather technical. Furthermore, I am not sure whether the method proposed in the paper is of practical relevance for researchers working with brain implants with a high density and number of neural probes. However, as the later point is not my core competence, I leave it up to others to judge on that.

Reviewer #3

(Remarks to the Author)

The revision has addressed most of my comments. However, there are still a few points I would like to highlight.

1. I find the work valuable and interesting, but, personally, it has left me with some unanswered questions. This work does not consider the crosstalk due to volume conduction based on the findings in 2.1. I believe this is the biggest weakness of the paper and my main concern regarding the analysis/outcomes. I would have liked this to have been addressed more in this paper, especially when the crosstalk due to volume conduction is more present than the crosstalk between the lines and what are the major contributing factors. The two points below are relevant to this.

a. I agree with the authors that, in practice, "the most realistic scenario is the one where two neighbouring electrodes are simultaneously recording". However, to study pure cross-talk (with the intent to understand the influence of the design/routing layout and interconnects) one should exclude any coupling of signal that comes through volume conduction (hence my former suggestion about adding an isolated electrode in the experiment and applying the correction algorithm on this electrode).

b. I am particularly missing a statement that the electrode impedance and insulation are apparently not the greatest contributors. Therefore, the electrode design itself should only be a secondary concern in the system design when considering cross-talk.

2. Regarding the use of electrode 1 (figure 3) or A (figure 8) to present and confirm the algorithm or hypothesis, it is still not clear to the reviewer: Figure 3, shows a high crosstalk correlation on electrodes 5,9,13 due to the signal coupling of electrode 1 (or A) to 5,9,13. Therefore, in Figure 8 c, the presentation of another electrode (5,9 or 13) would be better for a presentation of the algorithm since electrode A (or 1) contains the least cross talk since electrodes 5,9 and 13 carry the lowest amount of signal of interest.

3. The authors should add a note with the clarification that this study is a single recording session from a single implant.

Reviewer #4

(Remarks to the Author)

REBUTTAL LETTER

NCOMMS-24-14661A, *Bridging circuit modeling and signal analysis to understand the risk of crosstalk contamination in brain recordings*

Reviewer 1			
	Comment	Response	Location in Manuscript
1	The overall scope of the paper is highly technical. On one hand, I think the results are important because they systematically explore crosstalk contamination starting by discerning signals and understanding their correlation by modelling the various elements of the equivalent circuits in the signal chain. On the other hand, the orthogonality of the analysis based on different lumped elements, by simulating crosstalk and using the element model to propose a post-hoc correction algorithm, all validated against ECoG data makes the study highly technical.	Thank you for the insightful analysis. Our objective was indeed to combine a robust technical approach with in vivo electrophysiology experiments to effectively validate our technical work.	
2	Looking at the work critically, the algorithm operates under the hypothesis that it can accurately solve for the transfer functions between active and passive lines. This is based on the extension of the two-line transmission model to four-line, which is specific to the design of the routing in this implant. However, it would be beneficial to comment on how any given design of the implant can be extrapolated to obtain transfer functions for correction. Even better, it would help the audience to follow general design rules, which the authors should take the opportunity to propose to enable easy estimation of such correlations. Rules such as routing certain electrodes on the implant in a particular way and maintaining the same order of routing on connectors, PCBs, and so on.	We appreciate this comment and suggestion. While each system will have its unique routing and design, the methodology and tools we used to model our specific system can be applied to other systems as well. The specific simplifications, number of elements, or parameters might vary, but the underlying principles remain the same because the fundamental building blocks are the same. In our analysis, we focus on identifying the key elements that contribute most significantly to crosstalk. Rather than prescribing rigid design rules - which would be challenging to generalize and impractical given the need to optimize multiple parameters - we aim to help the readers understand how these elements interact and influence the transfer function. Our goal is to guide the development of future miniaturized implants by creating a better understanding of limiting factors, critical parameters, and opportunities for design optimization. For example, low impedance is a valuable metric for improving recording quality but also reducing the degree of crosstalk. Our primary recommendation is that "cross-checking recordings against the routing layout is imperative, as it can help discern and discard crosstalk contamination." This underlines that the ultimate quality check should rely on the recorded data itself. Incorporating baseline cross-correlation or coherence tests against the routing layout into the system characterization process is essential to rule out crosstalk contamination. We added a subsection to the discussion to clarify this point.	Discussion, subsection 3.3
3	Another clarification needed is on the validation of electrophysiological information in the MUA band, on the detection of spiking activity. The authors demonstrated that the number of spikes detected drops after correction due to a drop in amplitude. The authors hint that this is due to spurious detection of spikes from the original signal with crosstalk, but would it be possible that it could be an overcompensation from the algorithm potentially causing loss of relevant data? Maybe including additional criteria such as the refractory period in spike detection could accurately pinpoint to the nature of spikes? If the authors are indeed confident that these are glitches due to contamination, are there any suggestions to curb this?	We acknowledge the relevance of this comment. Validating the accuracy of corrected signals in terms of spike content would indeed require additional criteria, which are beyond the scope of this paper. The primary purpose of the crosstalk-correction algorithm presented here is to serve as a bridge between circuit modeling and real neural recordings, validating the hypothesis that crosstalk underlies the atypically high coherence levels observed between neighboring channels. Our analysis demonstrates that crosstalk contamination can be significant, with attenuation levels below -40 dB, enough to influence recordings in the high-frequency band. As a concrete example, we	Discussion, subsection 3.2

		show that simulated crosstalk at these levels can cause amplitude shifts in representative spikes beyond 1%. However, we emphasize that the aim is not to advocate for the use of correction algorithms as a remedy for low-quality recordings. In response to the reviewer's concerns, we have revisited the manuscript and revised some of our conclusions to provide more clarity and alignment with the feedback received.	
4	Do the authors have any underlying explanation for the dominant crosstalk contamination above 300 Hz and the increase in signal coupling with increasing frequency in the MUA band compared to the LFP band? It would be beneficial to address this in the discussion section.	It is expected that crosstalk would be more pronounced at higher frequencies, as capacitive crosstalk increases with frequency due to the reduction in capacitive impedance. We have added a paragraph to clarify to the broader readership the relation between frequency and crosstalk in the manuscript.	Introduction, paragraph 5
5	The study is performed on an acute implantation with all the impedance spectroscopies calculated around the same period. Could the authors please clarify the effect of ageing in the case of a chronic implant in terms of the change of dielectric properties of the insulation affecting both Z3 and Z4? In the simulation, the authors observe minimal variations in signal coupling with increase/decrease to Csh and Cimp but would Rsh or Rimp play any role?	Thank you for the interesting question. Two critical factors must be considered as an implant ages: changes in the electrode-tissue interface (Z2) and changes in the dielectric properties of the substrate material (Z3 and Z4). Regarding the latter, issues such as pinholes and delamination can have a significant impact on coupling between neighboring lines. In extreme cases, the insulation impedance could drop so low as to resemble a short circuit (a significant reduction in Rimp), or the shunt impedance could decrease dramatically, exposing a line to the conductive medium (a significant reduction in Rsh). However, these extreme scenarios involve resistive coupling overtaking capacitive coupling. To maintain focus, we chose to emphasize elements that most significantly affect the transfer function in well-performing recording systems, primarily the capacitive component. The capacitive component arising from the insulation material (polyimide in our case, though this may vary) may also vary over time due to water uptake in the polymer, which may impact the transfer function d. The rate of absorption varies substantially depending on the material, even within the same family of polymers. We have added a discussion of this consideration to the manuscript.	Discussion, subsection 3.1, paragraph 4
6	Figure. 1 could have the 'r' represented using a dotted line.	Thank you, we have edited Figure 1 according to your suggestion.	Figure 1
7	Figure. 7 has been mistakenly labelled as Figure. 6 (second time).	Thank you for noticing, we have fixed the mistake.	
8	Under methods – impedance spectroscopy – for Z4: the description reads 'One lead pair was connected to the left end of one track and the other to the right end of the second track' but in the supplementary figure. S1 shows both C and W extracted across the pins of the connector. Could the authors please clarify what is the left end and the right end per the figure?	Thanks again for noticing. We have fixed the issue.	Methods, subsection 8.6
	Reviewer 2		
	Comment	Response	Location in Manuscript
0	The main learning is that R_S and C_H dominate the cross-coupling. However, they are not coupling between the passive and active lines. Only the Z_n, n>2 components do that. Of course the authors also discuss the role of Z_5. However, this coupling is a 2nd order effect and I think that it can be minimized by the choice of proper connections. The coupling though R_S and C_H is no undesired coupling.	We fully agree with the reviewer's statement that R_S and C_H are not "undesired couplings," and it was not our intention to suggest otherwise. Our manuscript asserts that these elements are key determinants of the coupling observed between neighboring lines via Zimp and Zint. This assertion is based on the principle that the transfer function of a transmission line is significantly influenced by the termination impedance	Discussion, subsection 3.1, paragraph 2

	Instead, it is essential to create a map of brain activity.	at both ends: on the far side, by the amplifier's input impedance, and on the near side, by the impedance at the tissue-electrode interface. Specifically, for a given Z_{imp} and Z_{int} , a low-impedance electrode contact reduces the risk of crosstalk, while a high-impedance electrode contact amplifies the coupling into its respective line.	
1	Tests in Figure 2: Why do only electrodes 5,9, and 13 show strong correlation behavior and not e.g., 2, 3 and 6. How are the electrodes arranged on the micro-electrode disc?	This observation aligns with our crosstalk hypothesis. If electrodes 2 and 6 are physically closer to electrode 1 - and thus to the signal source, as proven by the higher spike count - why do they show a weaker correlation to electrode 1 compared to electrodes 9 and 13, which are further away? Examining the electrode arrangement (Figure 2a), it becomes clear that electrodes 9 and 13 share the same electrode column as electrode 1. According to our routing configuration (Figure 6b), this implies that these electrodes are closely wired together, thereby increasing the likelihood of crosstalk contamination. Indeed, this question underlines how hidden factors, such as routing, can contribute to crosstalk and influence the collected data in subtle, often overlooked ways. Through this work, we aim to raise awareness about undesired coupling originating from system components that are typically not considered in standard analyses.	Results, subsection 2.1, paragraph 4
2	What is the thickness of the 2nd PI layer, which is used to cover the Pt electrodes? Also $4\mu\text{m}$?	The thickness is $4\mu\text{m}$. We have clarified this in the recipe in the Methodology section.	Methods, subsection 8.1
3	Pt electrodes are commonly used for brain recordings. However, there are better electrode materials available and the authors should discuss the relevance of crosstalk for such electrodes. Furthermore, the authors do not comment on the performance of their Pt electrodes and how their transition impedance compares to the state-of-the-art. Crosstalk of electrodes is most relevant when the impedance of R_S and C_H are high (bad contacts). But they become negligible with good electrodes.	We fully agree that selecting the most suitable implantable material for a specific application is essential. Platinum (Pt) is one of the most commonly used electrode materials, making it a logical starting point for this discussion. While it is beyond the scope of this paper to experimentally evaluate multiple electrode materials, we acknowledge that electrode impedance can influence crosstalk and signal-to-noise ratio (SNR). We wish to highlight that, when the limitations of Pt electrodes are well understood and appropriately managed, successful outcomes can be achieved. These outcomes may be further enhanced by exploring additional electrode coatings, a topic that has been thoroughly investigated elsewhere (Lewis et al., 2024). In response to this comment, as well as feedback from Reviewer 1, we have expanded the discussion to clarify the role of electrode impedance and its potential impact on device performance. As noted in our earlier response to Reviewer 1, "Our goal is to guide the development of future miniaturized implants by fostering a deeper understanding of limiting factors, critical parameters, and opportunities for design optimization. For instance, impedance serves as a valuable metric for predicting both recording quality and the extent of crosstalk." We hope that our technical approach provides a useful framework to guide readers in selecting the most appropriate strategies and interpreting results effectively.	Discussion, subsection 3.3
4	The authors should provide fitting errors for their impedance analysis. In particular, I doubt that they can fit R_{IMP} and R_{int} in a meaningful way.	Thank you for suggesting it, we realize that it is valuable for the manuscript. To address this, we recalculated and re-fitted the values using an additional software to verify the consistency of the results. We have included a supplementary table demonstrating that the variation between the fittings (using two different software tools)	Supplemental information, Table S2

		is negligible. Our conclusions remain the same. Additionally, we observed, as anticipated, that the largest error corresponds to the resistive components of the model, while the capacitance-driven components can be fitted more accurately. These capacitance-driven components are the primary contributors to crosstalk analysis, which remains the focus of this study.	
5	As the authors discuss, variations in R_S , C_H and C_{int} have the largest influence on the impedance recording in the frequency range $<1\text{kHz}$. In particular, C_H can easily vary from experiment to experiment depending e.g., on the ion concentration in the biological environment and details of the electrodes. Due to the measured value of C_{int} , it also has a large influence on the reading. Although a 10x increase or decrease of C_{int} is theoretically possible, I doubt that this is going to happen in experiment. So, do you think that a 10fold variation of C_{int} is realistic in experiment? I would doubt that.	The figure aims to illustrate how different elements influence the transfer function. To emphasize these effects, we intentionally included extreme values, but these values are not meant to represent realistic conditions. Nevertheless, it is worth noting that liquid creep within connectors and PCBs, which can significantly alter impedance, is not uncommon. For consistency, we applied the same scaling factor across all elements. For components like electrode impedance, a 10-fold variation is entirely plausible, for instance, as a result of gliosis.	
6	Furthermore, combining the finding that only the MUA range is affected by x-talk and not the LFP (Figure 2), I would conclude that R_S and C_H are most important for coupling and not C_{int} (at it would also influence the LFP according to the spectra in Figure 5). So, following the impedance spectroscopy analysis, the coupling components Z_n , $n>2$ can be disregarded as they are typically at least 10x higher in impedance than Z_n , $n<3$. Which means that there is not coupling between V_{ACT} and V_{PAS} ?	Crosstalk between neighboring lines occurs due to coupling between V_{ACT} and V_{PAS} through Z_{imp} and Z_{int} , both of which consist of resistive and capacitive components. Since capacitive impedance decreases with frequency, this explains why crosstalk increases at higher frequencies (e.g., affecting the MUA band but not the LFP band). However, the coupling is influenced not only by the capacitance between the lines but also, to varying degrees, by all elements of the transmission line, including Z_n for $n<3$. We have provided additional details in our response to the first comment but are happy to discuss further if needed. We understand your position and acknowledge the complexity of this phenomenon. We hope this manuscript contributes to shedding light on it.	Discussion, subsection 3.1, paragraph 2
Reviewer 3			
	Comment	Response	Location in Manuscript
1	Crosstalk in neural recordings has been investigated in other publications. A simplified scheme which models the crosstalk from a brain-to-amplifier chain was already proposed by Najafi et al. (1990) and further developed in Du et al. (2009) and, in Seidl et al. (2012). Crosstalk in neural electrodes has been investigated without an electric circuit model by Qiang et al. (2021) (cited by authors) and Naughton et al. (2022). I suggest that the authors refer back to some of these relevant works and, importantly the original article where the model was first presented.	Thank you for your valuable feedback. We have revised the text to better highlight the connection between existing crosstalk studies in the literature and our proposed model. We recognize the importance of referencing prior work, both for the reviewer and potential readers, as it establishes a clearer context for the need to develop advanced models to characterize crosstalk and to provide specialized tools for crosstalk signal back-correction.	Introduction, paragraph 3
2	Overall, the authors present a thoroughly conducted series of experiments that confirm their claims. No additional evidence needed. However, it would be useful to see the level of crosstalk contamination after each stage in the simulations. For example, how much does the spread resistance really contribute?	Crosstalk is a cumulative phenomenon defined by the transfer function of the transmission line, which includes all components from the termination impedance at the proximal end (tissue-electrode interface) to the termination impedance at the distal end (amplifier input). To ensure the most accurate analysis, we modeled the recording system holistically, considering all its building blocks. Using simulations, we systematically varied one element at a time to evaluate its direct impact on the output signal in response to a known input signal (the influence of the spread resistance is illustrated in Figure 5). Analyzing the system at intermediate stages, rather than as an integrated whole, would contradict the foundational principles of our study and potentially yield misleading conclusions.	Discussion, subsection 3.1, paragraph 2

3	In figure 6, and section 2.5. electrode A (reference) is used as an example. Electrode A is indicated as the active electrode. A is also used as an example in Figure 6 c) for the correction algorithm. This indicates there is crosstalk from the electrode (B,C,D) (or all) to A. All other electrodes but A would be a more suitable electrode to analyse.	The crosstalk back-correction assumes that crosstalk happens within the sets of 4 channels composing each column of the grid. The mathematical model used for the correction is a system of 4 equations accounting for all coupling combinations among the 4 channels (A, B, C and D). We use electrode A as a reference electrode for the computation of coherence as this is the best representative example of an active electrode, as it records directly at the signal source (barrel C2), just as electrode 1 is used as reference for signal analysis in figure 2.	System of equations 2.1-4 Figure 7
4	Also waveform cross-correlations, as in section 2.1 (Figure 2.), should be reported. The same analysis for channels B,C, and D should be reported in the supplementary.	The focus on coherence plots is intentional, as the correction algorithm operates in the frequency domain. Its purpose is to prove that crosstalk affects the recorded signals, rather than to serve as a tool for "accepting" and then fixing crosstalk. Repeating the full analysis of Figure 2 on the corrected dataset would require additional validation and proof, which is beyond the scope of this paper.	
5	Additionally, how significant is the reduction in the coherence maps with the algorithm? Please include some statistics on that.	The level of significance depends entirely on the type of experiment and the specific information being extrapolated from it (e.g., the extent to which the study relies on coherence). The algorithm as such does only correct influence due to crosstalk but does not influence the level of coherence as outcome of neuroscientific hypotheses.	
6	In general, including in the experiment an isolated electrode (no opening) as a neighbour, would have given a more accurate representation of cross-talk coming from adjacent active lines/electrodes/interconnects.	This aspect is implicitly included in our analysis. An isolated electrode corresponds to a scenario where the termination impedance at the near end is infinite, which would amplify crosstalk coupling to that line. However, we disagree with the comment that this approach "would have given a more accurate representation of crosstalk." The most realistic scenario is the one where two neighboring electrodes are simultaneously recording. Termination impedance significantly influences crosstalk and isolation is not a working condition in well functioning electrode arrays.	Figure 6 Discussion, subsection 3.1, paragraph 4
7	Some fundamental theoretical background crosstalk principles and analysis is missing. E.g., it is known and expected that coupling increases with frequency. Could the authors please add this frame in the paper so that the reader can better appreciate and by extension replicate such analysis for a different setup?	Please see our response to reviewer 1 and 2, for convenience repeated here: "It is expected that crosstalk would be more pronounced at higher frequencies, as capacitive crosstalk increases with frequency due to the reduction in capacitive impedance." We acknowledge the need for clarification and added a similar paragraph in the manuscript.	Introduction, paragraph 5 Discussion subsection 3.3.
8	The algorithms for coherence maps and correlograms of spikes with the Matlab code and data should be available.	Thanks for the suggestion. We now specified all built in functions from NeuroExplorer and MATLAB used in the analysis in the methodology section.	Methods, subsection 8.5
9	Statement on page 19., section 3.1 Paragraph 1: "However, it is not intended as a method to surpass crosstalk contamination a posteriori. On the contrary, it should raise awareness to mitigate it at its source." Can you elaborate on that, what are the limitations?	We have touched on this in previous responses (Reviewer 1, comment 2) and edited the manuscript accordingly. 'In our analysis, we focus on identifying the key elements that contribute most significantly to crosstalk. Rather than prescribing rigid design rules - which would be challenging to generalize and impractical given the need to optimize multiple parameters - we aim to help the readers understand how these elements interact and influence the transfer function. Our goal is to guide the development of future miniaturized implants by creating a better understanding of limiting factors, critical parameters, and opportunities for design optimization. For example, low impedance is a valuable metric for improving recording quality but also reducing the degree of crosstalk.'	Discussion subsection 3.3.
10	Statement on page 20, the authors claim to crosscheck crosstalk against the routing layout: "Therefore, cross-checking the recordings against	We recommend always including baseline cross-correlation and/or coherence as part of system characterization. If the routing layout is reflected in the	Discussion, subsection 3.3

	the routing layout becomes imperative, as it can help discern and discard crosstalk contamination." How do the authors propose to address this quantitatively in a realistic scenario?	plots (as observed in our analysis), crosstalk cannot be ruled out, and the collective data may deviate from the ground truth. Identifying and isolating non-physiological patterns is essential for accurate data interpretation.	
11	Do electrodes A,B,C,D in figure 6 (by the way there are 2 figures numbered 6, please correct numbering) represent electrode 1,5,10,13 in figure 2, please clarify.	Thank you, we corrected the mistake. Channels A, B, C and D are a generalization for the channels composing any column of the grid. In Figure 7, we consider the first column of the grid as a representative case for analysis, meaning that A, B, C and D match electrodes 1, 5, 9 and 13 in Figure 3.	
12	Ground electrodes and reference electrodes are common mistakes, and in this scenario, the role of a ground electrode and reference electrode should be explained. Especially since the term reference electrode is used interchangeably. As stated on page 24: " Furthermore, an on-site reference electrode is included for bipolar recording configuration (not used for this study)." Furthermore, on page 25: " Using unipolar recording configuration, reference and ground pins of the headstage were tied together and connected to a skull screw." and on page 16: "Taking the original signals, coherence plots are computed from 3 Hz to 4 kHz for all sixteen electrodes using electrode A as reference". The model, therefore, is connected to one ground, so the brain is grounded to the same ground as the amplifier and the actual reference electrode is connected to the ground. It should be stated that the actual reference/ground electrode is not electrode 1 (A or reference) and that the actual reference/ground electrode placing has an impact on the recorded signal also related to crosstalk.	We fully agree with the reviewer and appreciate the need for clarification. Throughout the Results, Discussion and Conclusion sections, the term 'reference' is used exclusively to refer to the channel against which the correlation or coherence is computed. Only in the Methods section, under specifically contextualized subsections, is the term 'reference' used to differentiate reference and ground electrodes in a recording configuration or to differentiate reference, counter and working electrodes in an electrochemical setup. We have added a clarification on terminology as the first paragraph of the Results section.	Results, paragraph 1
13	Page 4, paragraph 1: on filling or to fill.	Thanks for pointing it out. We have fixed the issue.	
14	Figure 1, c: Please increase waveforms and amplitudes, maybe include the uncertainties in the cross-correlations.	There is very limited space in the panel to make waveforms and amplitudes larger. Our intent was to have a quick visual of the overall trend and show how SEP and spike amplitudes respect the distance to the signal source: the smaller the distance to the source, the higher the amplitudes. Nonetheless, we understand the need to show the data comprehensively and have added a figure in supplemental material where the same data is shown in detail. Regarding the uncertainties, we thank the reviewer for the suggestion. However, the crosscorrelation was plotted without an uncertainty measure since it was not calculated as a mean of several trials or different implants. Indeed, the calculation is rather a single example, made on the spike events detected from one recording session performed with one implant.	Supplemental information, Figure S1
15	Figure 6 occurs twice.	Thank you, we renamed the affected figures.	
16	Methods section 8.6: What are low force and low sense leads?	This nomenclature is standard for 4-point measurements. Two leads connect to the current source (high and low force), while two connect to the voltmeter (high and low sense). By shorting the high force with the high sense lead and the low force with the low sense lead, the setup effectively functions as a 2-point measurement. We have added this clarification in the manuscript.	Methods, subsection 8.6

REBUTTAL LETTER, FINAL COMMENTS BY REV. 3

NCOMMS-24-14661A, *Bridging circuit modeling and signal analysis to understand the risk of crosstalk contamination in brain recordings*

The revision has addressed most of my comments. However, there are still a few points I would like to highlight. I find the work valuable and interesting, but, personally, it has left me with some unanswered questions.

Thank you very much. We try to answer your questions and clarify why we have stuck to definitions that are common in the field from an engineering perspective.

1. This work does not consider the crosstalk due to volume conduction based on the findings in 2.1. I believe this is the biggest weakness of the paper and my main concern regarding the analysis/outcomes. I would have liked this to have been addressed more in this paper, especially when the crosstalk due to volume conduction is more present than the crosstalk between the lines and what are the major contributing factors.

The authors agree that volume conduction is a serious topic in neural signal discrimination. However, we would like to stress out that “crosstalk” and “volume conduction” are two separate phenomena and should not get convoluted, even though the outcome of both phenomena is a distorted signal (section 2.1. Discerning crosstalk from volume conduction addresses this distinction). Therefore, we cannot agree with the statement that volume conduction should get included in the crosstalk study.

Our study addresses solely the research question whether we can identify distortions due to limited insulation resistance in our technical setup in the electrophysiological recordings and can correct them. The question how volume conductivity in biological tissue influences the shape and frequency component from the signal source to the transducer and how superposition of spatio-temporally distributed signals sources in such an environment works and whether single sources can be separated is a very exciting one but not the topic of this study.

We believe we can solve this misunderstanding by adding the following text in the manuscript (line 150):

“It is important to highlight that the focus of this study is on the research question whether distortions in the electrophysiological recordings due to limited insulation resistance in the technical setup can be identified. The question of how volume conduction in biological tissues influences the signal shape and frequency component from the source to the transducer and how superposition of spatio-temporally distributed signal sources in such an environment works is not the aim of this investigation.”

The two points below are relevant to this.

a. I agree with the authors that, in practice, “the most realistic scenario is the one where two neighbouring electrodes are simultaneously recording”. However, to study pure cross-talk (with the intent to understand the influence of the design/routing layout and interconnects) one should exclude any coupling of signal that comes through volume conduction (hence my former suggestion about adding an isolated electrode in the experiment and applying the correction algorithm on this electrode).

We believe that - according to our understanding from the electrical engineering point of view - we did exclude volume conduction.

Prerequisite: Signal properties are seen independent from their origin here. Volume conduction delivers the kind of signal that we pick up. This signal will then be subjected to crosstalk through all stages through the signal chain (=the recording system). Figure 1 summarizes this concept, where a mathematical description of the true vs recorded signal is given.

Our measurements: We have performed the characterization of our components of this signal chain (as depicted in Figure 4) independent of the recorded signals and identified the electrical properties as parameters (see Figure 5 and paragraph 8.6) of the single components to combine them in a mathematical transfer model. The inverse of this mathematical model is then used to get rid of the signal distortions based on these electrical properties in between the different lines and pins (paragraph 2.4 and Figure 7). Thus, the characterization does not rely solely on in vivo recordings but on electrical properties extracted from separate experiments.

In conclusion, volume conduction is not taken into consideration as part of crosstalk in the whole approach. The authors therefore disagree with the suggestion to use an insulated electrode as reference for correction does improve our approach. This would make sense if an average signal from volume conduction should be subtracted as a kind of “common ground” which is not the intention of our study.

b. I am particularly missing a statement that the electrode impedance and insulation are apparently not the greatest contributors. Therefore, the electrode design itself should only be a secondary concern in the system design when considering cross-talk.

Electrode impedance is, in fact, concluded to be one of the main determining parameters to crosstalk in our case study (along with impedance at interconnects level, as per section 3.1). The electrode impedance as such acts as “termination impedance” of the signal chain. Changes in this value influence any other value in the signal processing chain.

The insulation between the adjacent lines of the substrate of the electrode array is a component of the signal transmission chain that should not be neglected at all according to our opinion. If the insulation strength (= resistance) is reduced due to flaws in manufacturing or damage during handling, interference from one line to another changes due to changes in the electrical properties (resistance and capacitance). All aspects, even if not the greatest contributors, are influencing the final crosstalk outcome. Besides, what is a determinant factor in one recording system, might be different in another, thus the importance of using the tools presented in this work to characterize each system (each will have a unique transfer function). Therefore, the recommended statement would be rather misleading than helpful to the reader. We emphasize that the electrode design definitely influences crosstalk parameters when we consider distance of lines as well as their width and relative arrangement to each other. Depending on how much water uptake and ion penetration come with the selected substrate and insulation material, strong differences in interferences (=crosstalk) between the lines will occur.

2. Regarding the use of electrode 1 (figure 3) or A (figure 8) to present and confirm the algorithm or hypothesis, it is still not clear to the reviewer: Figure 3, shows a high crosstalk correlation on electrodes 5,9,13 due to the signal coupling of electrode 1 (or A) to 5,9,13. Therefore, in Figure 8 c, the presentation of another electrode (5,9 or 13) would be better for a presentation of the algorithm since electrode A (or 1) contains the least cross talk since electrodes 5,9 and 13 carry the lowest amount of signal of interest.

We see your point and added a sentence to prevent confusion. Furthermore, we have repeated the analysis for a group of four channels A, B, C and D and show the complete coherence matrix and comparison of all high-pass filtered traces in the supplementary material.

We rephrased/added in line 343:

“The present results suggest that, under the boundary conditions specific to this recording system, crosstalk contamination is strong enough to cause spurious detection of spikes in the active channel recording the closest from the signal source. Furthermore, it is important to note that, generally, the signals recorded by the passive channels, even though weaker in terms of information content, are proportionally more affected by crosstalk contamination precisely due to the smaller amplitude signals (figures S2 and S3 in the supplementary material).”

And added figures S2 and S3 to the supplementary material:

Figure S2: Coherence matrix before and after crosstalk back-correction. Coherence computed for a representative set of four channels A, B, C and D composing an electrode column of the electrode grid of one implanted device, taking each of the four electrodes as the reference. Signals after correction are symbolized by an asterisk (A*, B*, C* and D*).

Figure S3: High-pass filtered signals before and after crosstalk back-correction. High-pass filtered (HPF) signals ($f_c = 300$ Hz) computed for a representative set of four channels A, B, C and D composing an electrode column of the electrode grid of one implanted device. Signals after correction are symbolized by an asterisk (A*, B*, C* and D*).

Further clarifications:

The back-correction algorithm takes the four channels A, B, C and D and computes the corrected signals using all channels against all others (system of equations 2.1-4), irrespective of what each is recording (the algorithm treats all channels the same, no matter how much signal of interest they carry).

Regarding figure 3, electrodes 5,9 and 13 have increasing distance to electrode 1 which is the closest to the barrel field from which the neural activity (moving the rat's whiskers) is recorded. The LFP activity (Fig 3e) decreases as expected since the distance increases. The magnitude of the MUA activity (Fig 3f), however, is higher than expected. This was surprising from a neuroscientific perspective and raised the question, if there could be some interference (=crosstalk) in the system that is the origin of this high signal strength. When we compare the values of electrodes 5 and 2 or 6, 9 and 7, 13 and 11, the magnitudes of 5,9, 13 are much higher as those of 2/6, 7 and 11, even though they are in similar distance to the barrel field. This was our starting point to model the transfer function of the system based on its electrical properties and apply the inverse of this transfer function to the signal and see what comes out (shown in figure 8); A=1, B=5, C=9, D=13. Even though channel 1(A) shows strong signals and is closest to the recording target, crosstalk even affects this channel in the MUA range by "higher frequency" crosstalk of the MUA signal components. LFP is not influenced here.

3. The authors should add a note with the clarification that this study is a single recording session from a single implant.

We would like to point out that we have performed experiments on four animals with four implants. Therefore, we kindly disagree and hope the above is clear from the methods section: 8.3 Animals and implantation procedure. Differences between four implants are shown, for example, in Figure 3 e,f.